# Somatically hypermutated antibodies isolated from SARS-CoV-2 Delta infected patients cross-neutralize heterologous variants

Haisheng Yu [1,12] ✉, Banghui Liu[2,12], Yudi Zhang[2,3,12], Xijie Gao [4,12], Qian Wang[5,12], Haitao Xiang[6,7,12], Xiaofang Peng[8,12], Caixia Xie[6,7,12], Yaping Wang[1,12], Peiyu Hu[4,9], Jingrong Shi [1], Quan Shi[6,7], Pingqian Zheng[4,9], Chengqian Feng[1], Guofang Tang[1], Xiaopan Liu[6,7], Liliangzi Guo[1], Xiumei Lin[6,7], Jiaojiao Li[1], Chuanyu Liu [6,7], Yaling Huang[6,7], Naibo Yang[6,7], Qiuluan Chen[4], Zimu Li[2], Mengzhen Su[2,10], Qihong Yan[2,5], Rongjuan Pei[11], Xinwen Chen [9,11], Longqi Liu [6,7], Fengyu Hu[1], Dan Liang[8], Bixia Ke[8], Changwen Ke[8] ✉, Feng Li [1] ✉, Jun He [2,4], Meiniang Wang [6,7] ✉, Ling Chen [1,2,4,9] ✉, Xiaoli Xiong [2,4] ✉ & Xiaoping Tang [1,9] ✉

SARS-CoV-2 Omicron variants feature highly mutated spike proteins with extraordinary abilities in evading antibodies isolated earlier in the pandemic. Investigation of memory B cells from patients primarily with breakthrough infections with the Delta variant enables isolation of a number of neutralizing antibodies cross-reactive to heterologous variants of concern (VOCs) including Omicron variants (BA.1-BA.4). Structural studies identify altered complementarity determining region (CDR) amino acids and highly unusual heavy chain CDR2 insertions respectively in two representative cross-neutralizing antibodies—YB9-258 and YB13-292. These features are putatively introduced by somatic hypermutation and they are heavily involved in epitope recognition to broaden neutralization breadth. Previously, insertions/deletions were rarely reported for antiviral antibodies except for those induced by HIV-1 chronic infections. These data provide molecular mechanisms for cross-neutralization of heterologous SARS-CoV-2 variants by antibodies isolated from Delta variant infected patients with implications for future vaccination strategy.

Evasion of vaccine-induced neutralizing antibodies is believed to be one contributing factor of breakthrough infections caused by SARS-CoV-2 variants[1]. Omicron variants effectively evade most characterized antibodies[2] likely contributing to an increased rate of breakthrough infections[3]. Studies of antibody responses in naive individuals infected by the original SARS-CoV-2 strain revealed that germline-like antibodies are widely and rapidly induced[4–6]. Antibody isolation and informatics studies revealed that germline antibodies of different clonotypes preferentially target specific spike protein surfaces generating antibodies of 4 different classes as define by epitope locations[7].

A full list of affiliations appears at the end of the paper. ✉e-mail: yuhaisheng@gzhmu.edu.cn; kecw1965@aliyun.com; gz8h_lifeng@126.com; he_jun@gibh.ac.cn; wangmeiniang@genomics.cn; chen_ling@gibh.ac.cn; xiong_xiaoli@gibh.ac.cn; tangxp@gzhmu.edu.cn

For examples, IGHV(VH)3-53/VH3-66 and VH1-2 germline antibodies are predominantly "class 1" and "class 2" antibodies respectively[8,9]. We recently identified that characterized VH1-69 antibodies are predominantly "class 2" antibodies[10]. The Omicron BA.1 variant emerged with a highly mutated spike protein bearing more than 30 substitutions with 15 located in the receptor binding domain (RBD) alone. Some of the RBD substitutions have been repeatedly observed in many variants, among these, substitutions at 417 and 484 are known to be highly efficient in evading currently isolated germline "class 1" and "class 2" antibodies respectively[11]. It has been reported that repeated exposures of SARS-CoV-2 antigens by either infections or vaccinations elicit superior humoral responses with neutralizing activities against the highly mutated Omicron BA.1 variant[12,13], however, the molecular mechanisms underlaying such superior immune responses remained incompletely characterized.

In this study, SARS-CoV-2 antigen specific memory B cells isolated from patients primarily with breakthrough infections of Delta variant are shown to exhibit higher level of somatic hypermutation (SHM). We identify a number of cross-reactive neutralizing antibodies (nAbs), with neutralizing activities towards wildtype (WT), Beta and Delta strains. In addition, a subset of these antibodies maintains neutralization towards the Omicron BA.1 variant. Two representative Omicron BA.1 variant neutralizing antibodies YB9-258 and YB13-292, encoded by commonly induced VH3-53 and VH3-21 antibody genes respectively, are identified as "class 1" and "class 2" RBD antibodies. These two antibodies feature residues introduced by somatic hypermutation or highly unusual heavy chain complementarity determining region 2 (HCDR2) loop insertions rendering them highly resistant to known RBD substitutions including substitutions at 417, 452, 484, and 501. By antibody informatics and structural analysis, we investigate structural and genetic basis of cross-neutralizing activities of YB9-258 and YB13-292. These data provide molecular mechanisms for cross-neutralization of SARS-CoV-2 variants by YB9-258 and YB13-292 isolated from Delta-infected patients, providing important data informing future vaccination strategy.

## Results

### High-throughput single B-cell sequencing revealed elevated SHM in Delta variant infection patients

Four pooled samples of peripheral blood mononuclear cells (PBMCs) were derived from blood samples of 15 COVID-19 convalescent patients primarily with breakthrough infections (13 have been confirmed vaccinated) of SARS-CoV-2 Delta variant (patient details are shown in Supplementary Table 1). Their plasmas show binding to various SARS-CoV-2 RBDs by ELISA (Supplementary Fig. 1). We isolated CD19$^+$ CD27$^+$ memory B cells which bind the homologous Delta variant RBD and S1 by FACS (Supplementary Fig. 2a) and subjected them to 10× Chromium 5'mRNA and V(D)J single-cell sequencing (Fig. 1a)[14]. After standard quality control, from the sorted CD19$^+$CD27$^+$RBD$^+$S1$^+$ B cells we obtained single-cell transcriptome data for 3286 CD19$^+$CD27$^+$RBD$^+$S1$^+$ B cells and 3554 single-cell V(D)J data for the same CD19$^+$CD27$^+$RBD$^+$S1$^+$ B-cell population. To characterize potential functional subtypes of the sorted B cells, we performed unsupervised clustering using the "Seurat" graph-based approach[15]. B cells are separated into 9 clusters and 4 major cell types are identified, including naïve B cells, non-switched memory B cells (non-switched MBCs), switched memory B cells (switched MBCs), and plasmablasts (Fig. 1b, c). As expected, transcriptome profiling reveal that memory B cells account for the vast majority (90.20%) of our FACS sorted B cells.

It has been reported that earlier SARS-CoV-2 infections elicit germline-like antibodies with lower levels of SHM, indicating an acute activation of naïve B cells. These studies were mostly based on non-vaccinated COVID-19 patients infected in initial stages of the pandemic[4–6]. However, comparing to a parallel single-cell sequencing dataset of 4,215 B cells derived from non-vaccinated COVID-19 patients infected with wildtype SARS-CoV-2 strain earlier in the pandemic (Supplementary Table 1), on average, we observed lower proportion of sorted B cells expressing unmutated VH genes in patients primarily with breakthrough infections (6.03 ± 0.74%) than in non-vaccinated patients (10.73 ± 1.26%) (Fig. 1d). Recent studies have reported recall of memory B cells in breakthrough infection rather than the activation of naïve B cells during primary SARS-CoV-2 infection reported in earlier studies[16–18]. We compared composition of antibody isotypes in selected B cells between the Delta variant infected and non-vaccinated COVID-19 patients. A 12% increase in the average percentage of isotype-switched B cells is observed in patients primarily with Delta variant breakthrough infections and significantly higher proportion of IGHG1 expression is observed (Fig. 1e and Supplementary Fig. 2b). Rapid induction of somatically hypermutated B cells and isotype-switched B cells likely suggests rapid recall of vaccine-induced memory B cells by breakthrough infections[16].

### Characteristics of mAbs isolated from Delta variant infection patients

We selected B cells with previously defined characteristics[14] to increase efficiency of identifying neutralizing mAbs. Briefly, we only selected clonotypes containing IgG1-expressing B cells with a somatic hypermutation rate higher than 2%. A total of 117 candidate mAbs (with an average VH gene SHM rate of 11.88% on nucleotide level) were selected, expressed and purified. Among them, 63 (with an average VH gene SHM rate of 9.63% on nucleotide level) can bind to RBD or S1 of Delta variant (Supplementary Dataset 1). To determine whether the 63 binding mAbs developed cross-variant activity, they were further tested binding to RBDs of SARS-CoV-2 wildtype, variants of concern (VOCs), variants of interest (VOIs), and SARS-CoV-1. 22 of the 63 mAbs show cross-binding to at least 5 of the VOCs and VOIs (Supplementary Dataset 2). Among them, 13 can bind to Omicron BA.1 variant RBD and 4 are cross-reactive with SARS-CoV-1 S1 (Fig. 1i and Supplementary Dataset 2). The 63 binding mAbs are distributed in 19 distinct VH gene families and showing high binding ratios in families such as IGHV2-5 (100.00%), IGHV3-66 (66.67%), IGHV3-53 (66.67%) and IGHV3-33 (66.67%) (Fig. 1f), recall of memory B cells expressing these VH genes has been observed in Omicron BA.1 breakthrough infections[16]. Comparisons between the binding mAb sequences and known mAb sequences reveal accumulation of somatic hypermutations in isolated binding mAbs (Fig. 1g). This is consistent with our observation that B cells from patients primarily with Delta variant breakthrough infections show sign of increased level of somatic hypermutation by single-cell V(D)J transcript data (Fig. 1d, g and Supplementary Fig. 2c, d). Interestingly, these cross-reactive mAbs are enriched in the C6 cluster, representing a highly specialized switched memory B-cell subpopulation (Fig. 1h and Supplementary Fig. 2e, f).

Guided by results of antigen binding (Fig. 1i and Supplementary Datasets 1 and 2), we tested neutralizing activities of 22 antibodies by either SARS-CoV-2 pseudoviruses or authentic viruses or both (Supplementary Dataset 2). Among the neutralizing mAbs, six of them show potent cross-neutralization against SARS-CoV-2 VOCs (with neutralization activities towards WT, Beta and Delta strains) with IC$_{50}$s lower than 0.05 mg/ml (of note, cross-neutralizing YB13-281 also binds SARS-CoV-1 S) (Fig. 2a). Interestingly, their binding activities are not completely correlating with their virus neutralizing activities. Among the 6 Omicron BA.1 spike binding antibodies, YB9-120 and YB13-208 bind wildtype and Delta spikes with high affinities and they exhibit reduced but still substantial binding towards the Omicron BA.1 spike (Fig. 2b). Surprisingly, they lose neutralizing activities completely towards Omicron BA.1 virus (Fig. 2a). It has been reported in other studies that binding activities are not always correlating with antibody virus neutralizing activities[19,20]. YB9-258 and YB13-292 demonstrate best cross-neutralization abilities towards VOCs, including the Omicron BA.1 variant (Fig. 2a). For these two antibodies YB13-292 has good

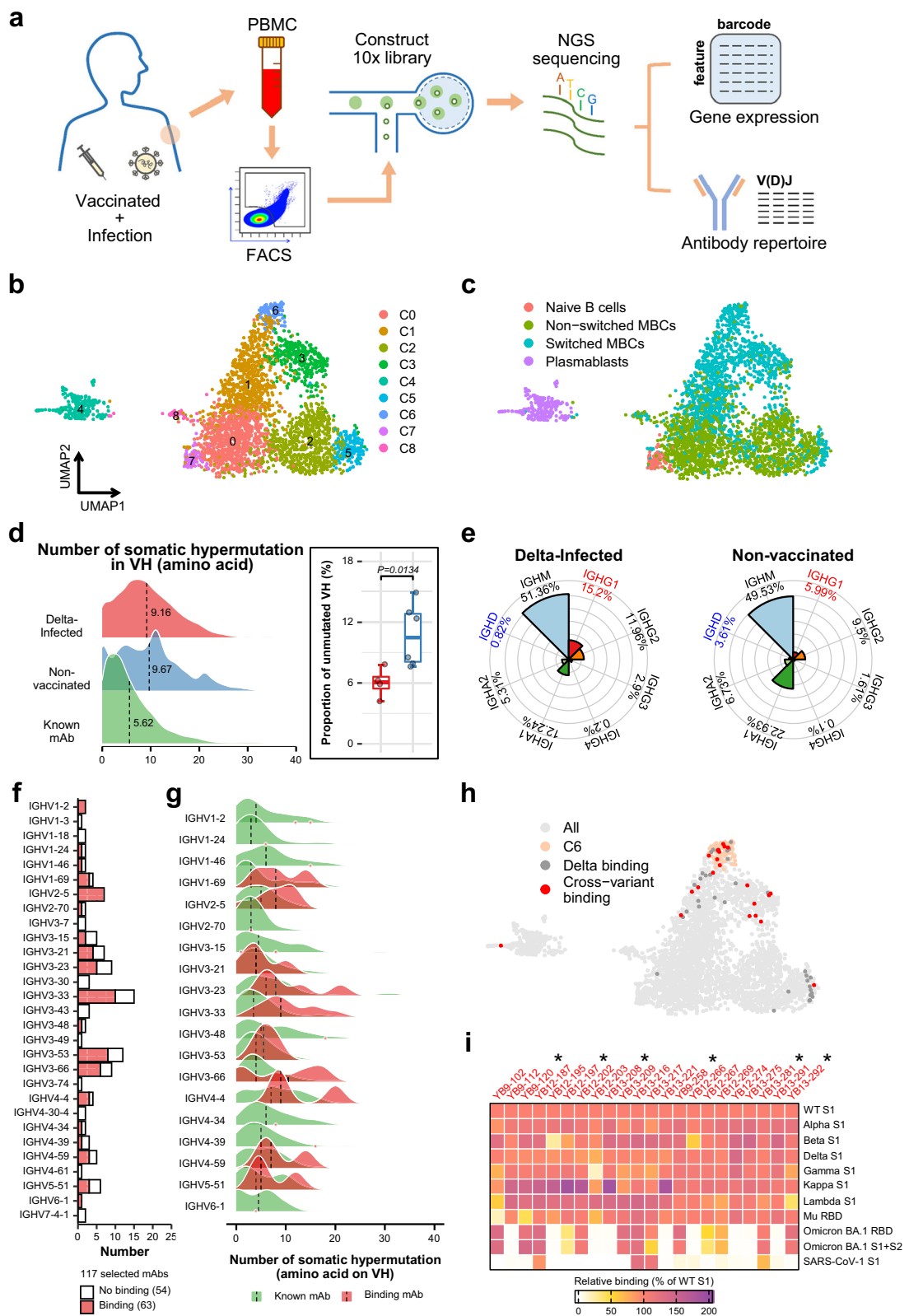

affinity towards wildtype, Delta and Omicron BA.1 spikes, while YB9-258 binding to wildtype, Delta and Omicron BA.1 spikes exhibits lower responses in BLI assays (Fig. 2b). Substitutions in Delta and Omicron BA.1 spikes appear to have little effect on YB9-258 and YB13-292 binding (Fig. 2b). We performed further BLI assays with RBDs of variants or RBDs bearing single substitutions to test their effect on YB9-258 and YB13-292 binding. We find that YB9-258 maintains binding to

all the tested variant RBDs and RBDs with single mutations (Fig. 2c and Supplementary Fig. 3a). While YB13-292 is resistant to most single RBD mutations that were previously known to be highly detrimental to binding of many known RBD antibodies[21] (Fig. 2c and Supplementary Fig. 3b). RBD substitutions in Kappa (L452R, E484Q) and Lambda (L452Q, F490S) variants abolish YB13-292 binding. To understand neutralizing activities of cross-neutralizing antibodies identified by our

**Fig. 1 | Single-cell atlas of memory B cells and identification of cross-neutralizing antibodies from patients primarily with Delta breakthrough infections. a** Overview of experimental design. CD27+ Delta-S1+ Delta-RBD+ B cells from patients primarily with Delta breakthrough infections were sorted by fluorescence-activated cell sorting (FACS) and subject to single-cell immune transcriptome (3286 cells, analyses shown in **b**, **c**, **h**) and V(D)J sequencing (3554 cells, analyses shown in **d**, **e**, **f**, **g**). **b** UMAP projection of B cells shows formation of 9 clusters. Each dot corresponds to a single-cell, colored according to identified clusters. **c** UMAP projection of B cells colored by the 4 major B-cell types, including naïve B cells, non-switched memory B cells (non-switched MBCs), switched memory B cells (switched MBCs), and plasmablasts. **d** Density plot showing SHM counts on variable heavy chain gene (VH) sequences of B cells from Delta-infected patients, non-vaccinated patients and antibodies from CoV-AbDab[69] (Known mAb). SHM count is defined by numbers of mismatched amino acids on VH using IgBlast. The mean values are indicated by dashed lines. The ratio of B cells with unmutated VH sequences is shown in the right panel. *P* value is calculated by two-tailed unpaired Student's *t*-test. The boxplots depict the median (horizontal line), upper/lower quartiles (boxes), and range (whiskers). **e** Fan charts comparing percentage of 8 antibody isotypes between Delta-infected patients and non-vaccinated patients. Isotypes with significantly higher or lower percentages in Delta-infected group are colored in red or blue, a result from Supplementary Fig. 2b. **f** Barplot showing VH gene usage frequencies of 117 mAbs selected from Delta-infected patients. **g** SHM compared between binding mAbs and known mAbs. SHM is measured as described in **d**. **h** UMAP projection of B cells expressing cross-reactive mAbs (red), binding mAbs (dark gray) and cluster C6 (pink). **i** Heat map showing normalized (to binding of wildtype S1) binding affinities of identified cross-reactive mAbs towards S1 of SARS-CoV-2 variants and SARS-CoV-1. Six potent cross-neutralizing mAbs (IC$_{50}$ < 0.05 mg/ml) are marked with *. Source data for **d**, **e**, **g** and **i** are provided as a Source Data file.

---

activity assays we determined the cryo-EM structures of the most potent YB9-258 and YB13-292 antibodies in complex with various spikes proteins.

## Residues introduced by SHM are heavily involved in epitope recognition by cross-neutralizing antibody YB9-258

We determined cryo-EM structures of antibody YB9-258 in complex with hexa-proline stabilized Omicron BA.1 spike (Omicron BA.1 S-6P) (Fig. 3a and Supplementary Figs. 4a and 5a). VH3-53 gene have been previously identified to encode a class of public antibodies binding to an epitope highly overlaps the ACE2 binding site[5,7,9,22]. They belong to the previously defined "class 1" RBD targeting antibody class with an epitope only accessible when RBD adopting an "up" conformation[7], notably this epitope has not been affected by the Delta variant RBD substitutions (L452R, T478K). We find that antibody YB9-258 engages the canonical epitope of "class 1" VH3-53 antibodies and consequently all YB9-258 bound RBD adopt an "up" conformation (Fig. 3a and Supplementary Figs. 4a and 5a). Although we used molar excess of YB9-258 Fab at high spike concentration (3.1 mg/ml, 22 μM protomer) during sample preparation, we only observed Omicron BA.1 spikes bound by 1 or 2 Fabs (Fig. 3a and Supplementary Figs. 4a and 5a). Analysis of these structures suggests that interactions between "up"–"down" RBDs likely hinder opening of the third RBD contributing to absence of 3:3 protomer:Fab complex in the cryo-EM sample (Supplementary Fig. 6a). We noted that similar interactions between "up"–"down" RBDs were observed when Omicron BA.1 spike bound to ACE2. These interactions were facilitated by hydrophobic surfaces exposed by Omicron BA.1 specific S371L, S373P, and S375F substitutions and were previously proposed to affect ACE2 binding[23–26] (Supplementary Fig. 6a).

YB9-258 bound Omicron BA.1 S-6P exhibits high structural dynamics preventing high-resolution structure determination (Supplementary Figs. 4a and 5a). In order to obtain more suitable samples for high-resolution cryo-EM structure determination, after extensive efforts, following a similar strategy previously described[26], we obtained high-resolution structures around the YB9-258 epitope using wildtype 6P spike (wildtype S-6P) incubated with YB9-258 Fab and an additional Fab of antibody R1-32 which we previously characterized[10] (Fig. 3b, c and Supplementary Fig. 4b). YB9-258 and R1-32 target distinct RBD areas and were able to bind an RBD simultaneously allowing formation of 3:3:3 YB9-258:R1-32:spike protomer structures (Fig. 3b and Supplementary Fig. 4b). Probably similar to other 3 RBD "up" spike structures previously characterized[10,27,28], this structure is highly labile and we observed many antibody bound S1 particles likely derived from disintegrated spikes (Fig. 3c and Supplementary Fig. 4b).

We obtained a 3.3 Å resolution structure of disintegrated S1 bound to YB9-258 and R1-32 (Fig. 3c and Supplementary Figs. 4b and 5d). In this complex, both YB9-258 and R1-32 bind to RBD surfaces that would be partially obstructed if the RBD is "down" within a spike trimer. We have previously found that antibody binding to such epitopes

can promote spike opening resulting in premature triggering or disintegration of spike trimer[10,27]. This structure also allowed us to model antibody–epitope interactions unambiguously to reveal that all CDRs of antibody YB9-258, except for LCDR2, contact the canonical "class 1" VH3-53 antibody epitope at the distal end of RBD (Fig. 3d and Supplementary Fig. 6). In this epitope, residue 417 is centrally located while residues 484, 452 and 490 are outside of the epitope. K417N has been usually found to completely abolish binding of "class 1" VH3-53 antibodies[9,11]. Although K417N only mildly affects YB9-258 binding but its effect is the strongest among common RBD single substitutions we tested (Fig. 2c, left panel). Consistently, YB9-258 exhibits reduced binding to RBDs of Beta and Omicron BA.1 variants both containing the K417N substitution, while RBDs with E484K/Q, L452R and F490S substitutions have little effect on YB9-258 binding. HCDRs and LCDRs of YB9-258 bury 694.9 A$^2$ and 533.5 A$^2$ of RBD surface areas respectively. Residues in CDRs mediate extensive contacts with the epitope, among these many are extra or altered hydrophobic residues introduced by somatic hypermutations (Fig. 3d and Supplementary Fig. 7). These include F27L in HCDR1, S53A and Y58F in HCDR2, S30G in LCDR1, and F94V, P95L in LCDR3 (Fig. 3d). Repeated antigen exposures due to vaccinations and infections likely activated affinity maturation which selected for antigen binding enhancing changes introduced by SHM. Comparison with two other isolated "germline-like" VH3-53/(IGKV) VK1-12 SARS-CoV-2 RBD antibodies (P4A1[29], LY-CoV481[30]) showed that, due to residues introduced by SHM in YB9-258 LCDR3 (F94V, P95L, and P96A) (Fig. 3d and Supplementary Figs. 6 and 7), LCDR3 of YB9-258 is adopting a different conformation to engage a closer contact with RBD surface (Supplementary Fig. 6b–e). Unlike most characterized VH3-53 antibodies which are highly susceptible to escape by the K417N change in spike RBD, YB9-258 is able to bind RBDs with K417 substitutions with small compromises in affinity (Fig. 2c and Supplementary Fig. 3a), and other common RBD substitutions were found not to affect YB9-258 binding. Consistent with the mutagenesis analysis, YB9-258 is able to bind Omicron BA.1 spike or RBD without significant change in affinity and maintains neutralization of Omicron BA.1 virus (Fig. 2a–c). By comparisons to VH3-53 antibodies known to be affected by K417 mutations, including B38[31], CC12.1[32], and CV30[33], we found that these antibodies bind K417 by cation-π interactions with conserved heavy chain residues Y33 and Y52. These interactions are often stabilized by salt bridges to negatively charged D or E in HCDR3 (Supplementary Fig. 6f–i). In contrast, the P95L change introduced by SHM in YB9-258 LCDR3 hinders K417 to form cation-π interactions with Y33 and Y52 (Fig. 3d and Supplementary Fig. 6b). Absence or hindrance of K417 interactions to Y33 and Y52 was also observed for mAbs 222[34], CS23[35] (Supplementary Fig. 6j–l). These antibodies are known to neutralize Beta variant with K417N/E484K/N501Y changes in RBD. But in the case of CS23, hindrance of K417 interactions by its HCDR3 residue M98 also abolished its ability to bind wildtype spike[35] in contrast to YB9-258 which retains wildtype spike binding. Based on the above mutagenesis

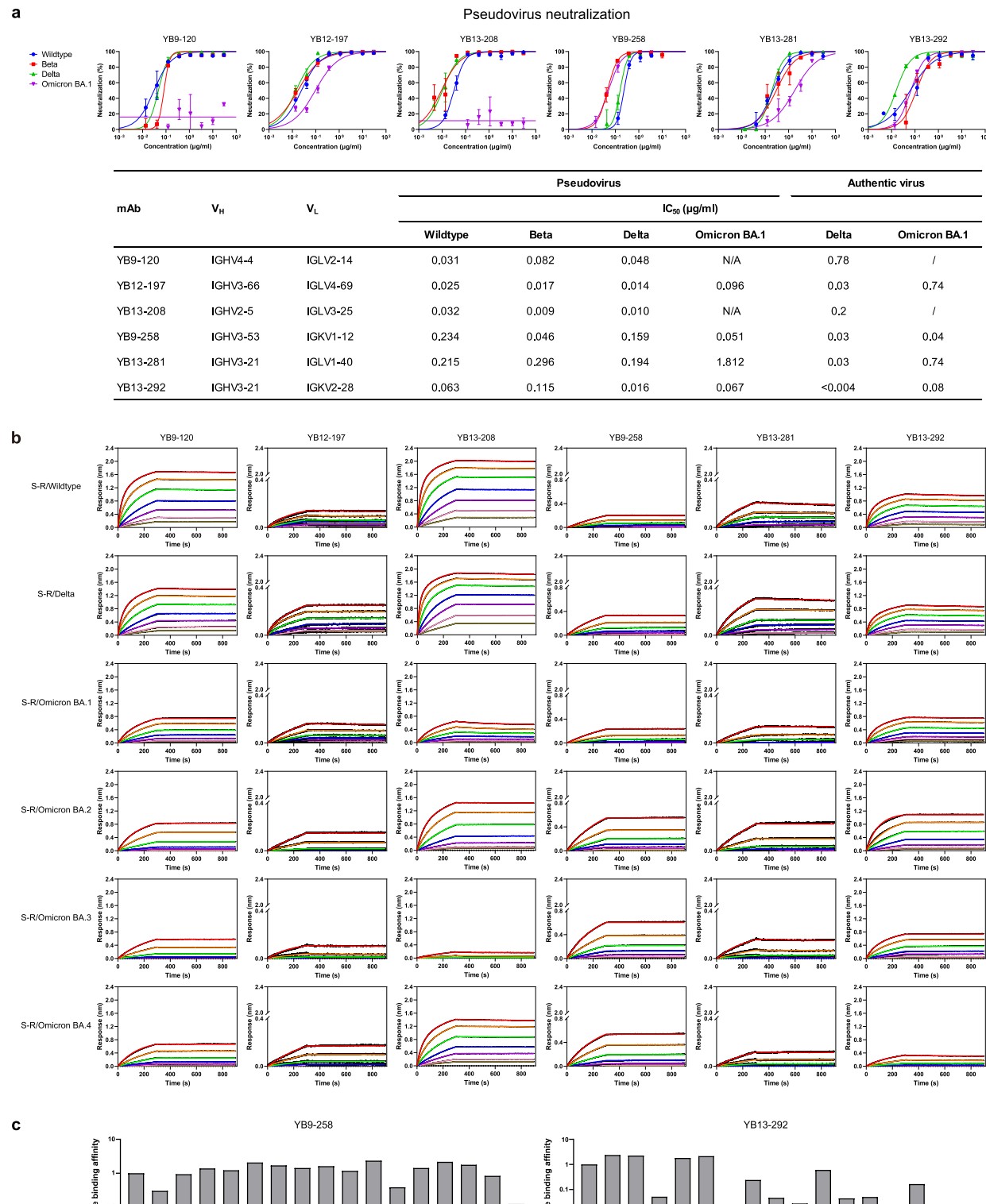

**Fig. 2 | Neutralization and binding activities of identified potent cross-neutralization antibodies. a** Neutralization activities of YB9-120, YB12-197, YB13-208, YB9-258, YB13-281, and YB13-292 IgGs towards wildtype, Beta, Delta and Omicron BA.1 pseudoviruses (data are presented as mean values ± SD). $n = 2$-3. Representative data from at least 2 independent experiments are shown. **b** Binding curves of YB9-120, YB12-197, YB13-208, YB9-258, YB13-281, and YB13-292 to wild-type, Delta and Omicron spikes are shown. Antibody binding was assessed by biolayer interferometry, IgGs were immobilized onto protein A sensors and binding to spike protein dilution series (200 to 3.125 nM) were recorded, binding kinetics parameters are summarized in Supplementary Table 2. **c** Effect of common SARS-CoV-2 RBD mutations on YB9-258 and YB13-292 binding. Fold changes are normalized to $K_D$s calculated from binding curves to wildtype RBD (RBD binding curves are shown in Supplementary Fig. 3a, b and $K_D$ data used for comparisons are shown in Supplementary Table 3). Source data for **a** are provided as a Source Data file.

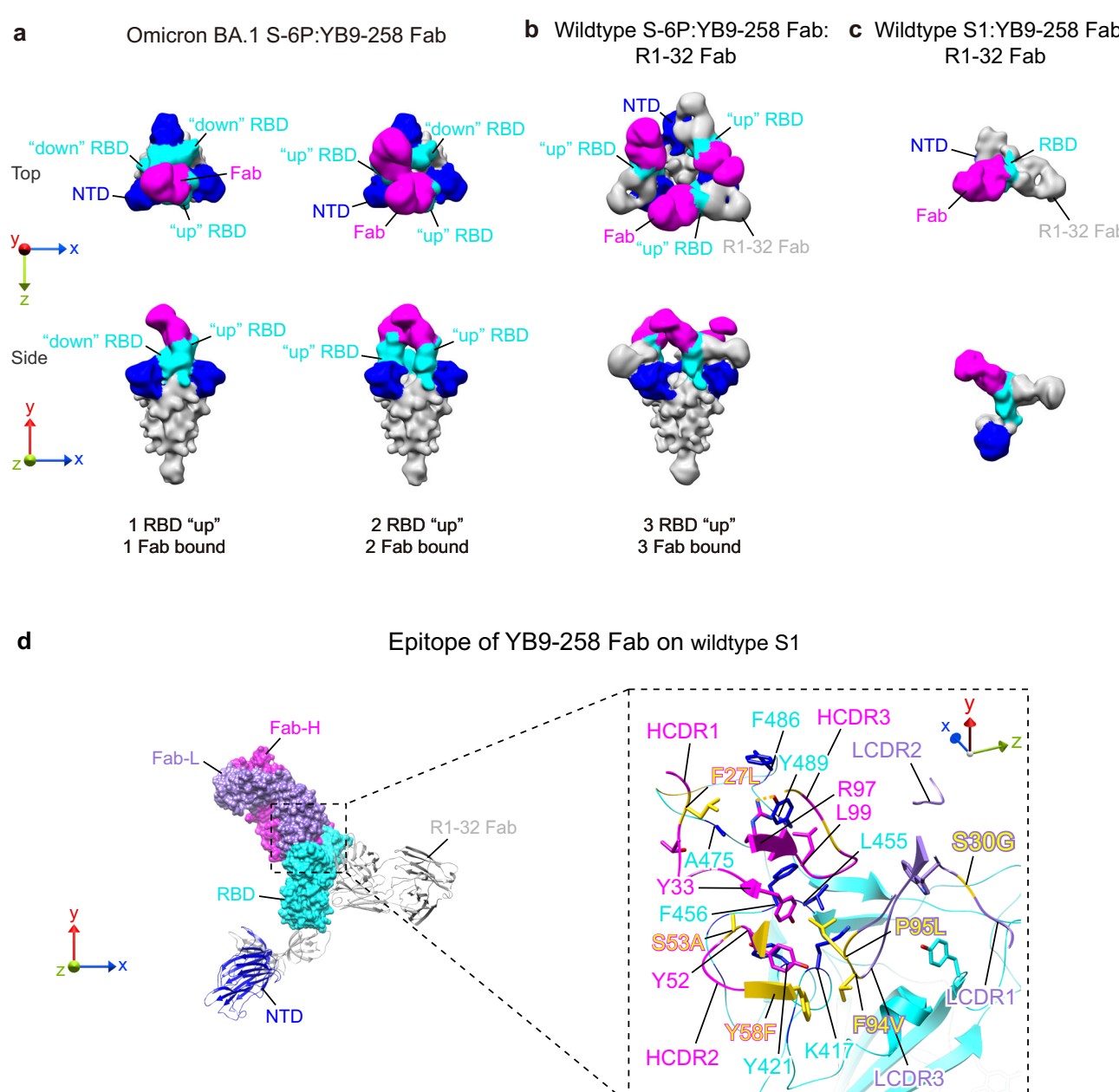

**Fig. 3 | Structure complexes formed between YB9-258 and SARS-CoV-2 spikes reveal an antibody-antigen interface involving point changes introduced by somatic hypermutation. a** Omicron BA.1 spike (S-6P) in complex with YB9-258 Fab in different stoichiometries and conformations. **b** Wildtype spike (S-6P) in complex with YB9-258 Fab and R1-32 Fab. **c** S1 of disintegrated wildtype spike in complex with YB9-258 Fab and R1-32 Fab. Structures in **a**–**c** are low-pass filtered to 15 Å to reveal flexible regions (also see Supplementary Fig. 5). **d** Epitope of YB9-258 Fab on RBD of wildtype spike. CDR loops are indicated, selected interacting residues between RBD and antibody are shown and indicated. Somatically hypermutated residues are highlighted in yellow and sidechains of the ones involved in interaction are shown with changes indicated (also see Supplementary Fig. 7). Dashed lines indicate hydrogen bonds. Highly buried RBD residues (BSA > 30 Å) are colored blue.

and structural analyses we conclude that affinity maturation likely selected SHM introduced changes which make YB9-258-RBD contact more extensive and less dependent on K417, rendering YB9-258 resistant to known RBD substitutions, allowing it to maintain neutralization of Omicron BA.1 variant.

## A highly unusual HCDR2 insertion is essential for cross-neutralizing activities of YB13-292

By cryo-EM, we only observed 3:3 spike protomer:YB13-292 Fab complexes (Fig. 4a and Supplementary Fig. 4c), suggesting that excess of YB13-292 Fab is able to saturate all three spike RBDs. Structures of YB13-292 bound to 0, 1 and 2 RBD "up" spikes (Omicron BA.1 S-6P) were

captured (Fig. 4a). These structures demonstrate that YB13-292 binds to an RBD outer surface which is accessible when RBD is in both "up" and "down" conformations. Aligning YB13-292 and germline IGHV3-21 sequences identifies a highly unusual 4 amino acid "SNIL" insertion into HCDR2 (Supplementary Fig. 7). The 3.8 Å resolution focused refined structure (Supplementary Figs. 4c and 5i) reveals that this insertion introduces hydrophobic residues $55I_{ins}$, $56L_{ins}$ at the tip of HCDR2 (Fig. 4b and Supplementary Fig. 8). The YB13-292 binding interface is dominated by residues putatively undergone SHM (Fig. 4b and Supplementary Fig. 7). In particular, we note that the VH1-69 "HCDR2 epitope", a hydrophobic patch in RBD formed by L452, F490 and L492 previously noted by us[10,36], is contacted by SHM introduced

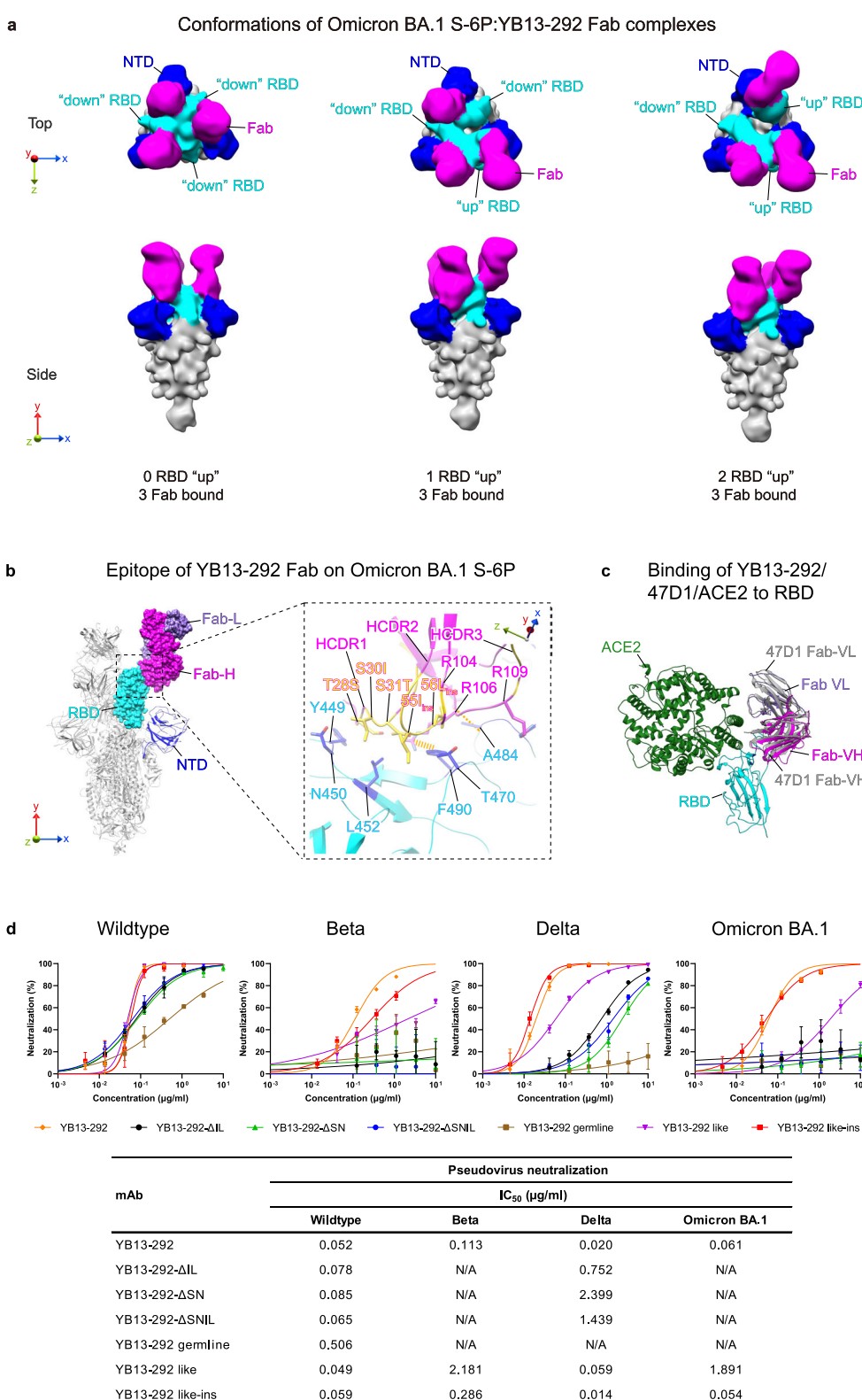

S30I of HCDR1, $55I_{ins}$ and $56L_{ins}$ of HCDR2 (Fig. 4b). We note that this recognition mode at RBD residue 452 is highly similar to previously characterized VH1-69 antibodies. Comparisons revealed that YB13-292 epitope and binding mode are highly similar to VH1-69 antibody 47D1[37] (Fig. 4c and Supplementary Fig. 8). Consistent with structural data, substitutions at 452, 484 and 490, all within the epitope, affect YB13-292 binding (Figs. 2c and 4b). Substitutions at 417 and 478, outside of the epitope, have very little effect (Figs. 2c and 4b). However, different

from 47D1, for which E484K and F490S are able to completely abolish binding, YB13-292 is able to maintain substantial binding to RBDs with E484K and F490S substitutions (Supplementary Fig. 3b, c). Modeling suggested that YB13-292 and ACE2 would not clash when bound to RBD (Fig. 4c). 47D1 has been shown to neutralize virus by inhibiting spike fusogenic activity without blocking ACE2 binding[37]. With almost the same epitope and binding mode to 47D1, YB13-292 likely neutralizes virus via the same mechanism as 47D1 (Fig. 4c).

**Fig. 4 | Structure complexes formed between YB13-292 and SARS-CoV-2 Omicron BA.1 spike reveal a functional "SNIL" insertion in HCDR2. a** YB13-292 Fab in complex with Omicron BA.1 spike (S-6P) in different stoichiometries and conformations (structures are low-pass filtered to 15 Å). **b** Epitope of YB13-292 Fab on RBD of Omicron BA.1 spike. Detailed interactions between YB13-292 Fab and Omicron BA.1 spike are shown in the dashed box. CDR loops are indicated, selected interacting residues in antibody-RBD interface are shown. Somatically hypermutated/inserted residues are highlighted in yellow and sidechains of the ones involved in interaction are shown with changes indicated (also see Supplementary Fig. 7). Thick and thin dashed lines indicate cation-π and hydrogen bond interactions respectively. Highly buried RBD residues (BSA > 30 Å) are colored blue. **c** Comparison of YB13-292, 47D1 and ACE2 binding to RBD. YB13-292 and ACE2 are shown in colors and 47D1 is shown in gray. **d** Different variants of YB13-292 antibody were generated, including YB13-292 without the "SNIL" insertion (YB13-292-ΔSNIL), or with variations of the "SNIL" insertion (YB13-292-ΔIL, YB13-292-ΔSN), and YB13-292 reverted to predicted germline sequence (YB13-292 germline). A "YB13-292 like" antibody was also identified from single-cell sequencing data and "SNIL" was inserted into its HCDR2 (YB13-292 like-ins) (see Supplementary Fig. 9 for details). Neutralization of wildtype, Beta, Delta, and Omicron BA.1 pseudoviruses by YB13-292 and YB13-292 variant IgGs (data are presented as mean values ± SD). $n = 2–4$. Representative data from at least 3 independent experiments are shown. $IC_{50}$ values are summarized in the table below. Source data for **d** are provided as a Source Data file.

In humans, germline IGHV1-69 genes are the only genes with a hydrophobic HCDR2 tip[36]. Hydrophobic HCDR2 loops have been shown to be probably the most important feature for VH1-69 antibodies, implicated in immunity against diverse viral pathogens[38]. Therefore, hydrophobic residues introduced by the "SNIL" insertion at the HCDR2 tip appear to confer VH3-21 antibody YB13-292 VH1-69 antibody characteristics, making its binding mode almost the same as VH1-69 antibody 47D1 and similar to other VH1-69 antibodies (Fig. 4b and Supplementary Fig. 8). Possibly due to the highly unusual elongated HCDR2 loop, YB13-292 binding buried the largest surface area at 452, and 6th largest area at 490 among structurally characterized antibodies contacting 452 (Supplementary Fig. 10). Both RBD residues 452 and 490 are known mutation hot spots[10,21,36], suggesting that antibodies recognizing this epitope pose strong immune pressure towards SARS-CoV-2. L452R featured in the Delta variant was shown to escape VH1-69 antibody LY-CoV555 targeting a similar epitope as YB13-292[39]. We performed mutagenesis to investigate effect of HCDR2 insertions on antibody function. We found that while the HCDR2 insertions are not essential in neutralization of wildtype virus, addition of the complete 4 amino acid "SNIL" insertion is essential for YB13-292 and YB13-292 like antibodies to fully neutralize Delta and Omicron BA.1 variants (Fig. 4d). These neutralization results are consistent with antigen-binding data (Supplementary Fig. 11), suggesting that the "SNIL" insertion was selected to overcome escape by variants.

In addition to hydrophobic contacts mediated by the "SNIL" insertion, binding of antibody YB13-292 is strengthened by additional interactions including a hydrogen bond between R104 of HCDR3 and backbone carbonyl oxygen of RBD residue 484, cation-π interaction between R106 of HCDR3 and F490, and further contact near A484 by HCDR3. Together with interactions mediated by the "SNIL" insertion, these additional interactions likely render YB13-292 resistant to escape: single substitutions at 484, 452 and 490 are unable to abolish YB13-292 binding with primarily effects on antigen dissociation (Fig. 2c and Supplementary Fig. 3b), explaining YB13-292's ability to cross-neutralize Beta, Delta and Omicron BA.1 variants (Fig. 2a). We found only simultaneous substitutions at 452/484 and 452/490 are able to abolish YB13-292 binding (Fig. 2c and Supplementary Fig. 3b), this is in contrast to 47D1 (Supplementary Fig. 3c) and previously characterized "class 2" antibodies that a single substitution at 484 was able to abolish their binding[10,11]. Again, analysis of YB13-292 binding reveals that residues introduced by SHM facilitate more extensive interactions between YB13-292 and RBD, conferring resistance to RBD substitutions.

## Discussion

By investigating antibodies isolated from patients primarily with breakthrough infections of Delta variant, we find that some binding antibodies isolated are able to cross-react to certain highly mutated Omicron variants. Previously, it has been identified that only 32 of the 247 previously isolated neutralizing antibodies retained neutralization of Omicron BA.1 pseudovirus[2]. We find antibodies isolated from patients primarily with Delta variant breakthrough infections generally possess more somatic hypermutations. Structural studies show that point changes introduced by SHM in YB9-258 are heavily involved in epitope recognition, likely rendering this antibody resistant to escape by 417 substitutions shown previously highly efficient in evading VH3-53/VH3-66 antibodies. In addition to SHM introduced amino acid changes, we also identify a functional insertion in the HCDR2 of YB13-292. We show that acquisition of this insertion confers an enhanced binding affinity and neutralization breadth for both YB13-292 and a YB13-292 like antibody. Probably due to that insertions or deletions (indels) account for only a small fraction of the somatic mutations in the normal human B-cell repertoire[40], insertions at HCDR2 are rarely reported in antiviral antibodies, except for those found for HIV-1[41]. Previously, a 3 amino acid insertion was identified near HCDR2 of 2D1, an influenza virus-specific antibody. This insertion strengthen binding to hemagglutinin (HA) indirectly by removing conflicting and unfavorable interactions[42]. Instead, the 4 amino acid "SNIL" insertion in YB13-292 is identified to directly mediate enhanced interactions with SARS-CoV-2 RBD. Possible interpretation for occurrence of these functional insertion/deletion is that, like point changes introduced by SHM, they are positively selected during stochastic B-cell differentiation in germinal centers for enhanced affinity[43,44]. A positive correlation was found between the occurrence of insertion/deletion and the degree of SHM in chronic HIV-1 infection[41]. This study also demonstrated an unusual high frequency of insertion/deletion among HIV-1 broadly neutralizing antibodies (bnAbs) and validated the critical role of indels for bnAb activity with an example of VRC01-like bnAb lineage. In addition to YB13-292, there is another VH3-21-encoded mAb isolated from this study, YB13-281, which displays cross-reactivity not only towards RBDs of SARS-CoV-2 variants but also SARS-CoV-1 RBD (Fig. 1i). Different from YB13-292, YB13-281 utilizes VL1-40 for light chain and this might be responsible for its unique activity. Overall, by investigating binding of two highly mutation resistant mAbs targeting distinct RBD epitopes, we identify that although both antibodies show similarities in binding modes compared to previously identified germline antibodies, amino acid substitutions and insertions introduced by SHM in these two mAbs enhance contact with epitopes. Previously, single mutations at 417 and 484 were known to be highly effective in evading "class 1" and "class 2" antibodies[9,11,45]. These single mutations are shown to be much less effective in abolishing binding of antibodies showing higher levels of SHM identified in this study. It has been observed that in SARS-CoV-2 convalescents, B cells undergo further affinity maturation, accumulate SHM, and express antibodies with improved potency and breadth[46–49]. It has also been shown that cross-reactive antibodies may be boosted by vaccine or infection[12,50–52]. A few structures of cross-reactive antibodies which retain activities against Omicron variants have been determined[53,54]. Among these, Sheward et al. reported a VH3-53 antibody with different CDRs from YB9-258 likely to withstand Omicron variants by a different molecular mechanism[54]. It has been reported that Omicron BA.1 resistant VH3-53/VH3-66 and VH1-69 antibodies are recalled in Omicron BA.1 breakthrough infections[12,16,36]. Isolation of Omicron resistant VH3-53 and related VH3-66 antibodies YB12-197 and YB9-258 in this study

highlights potential roles of mutation resistant VH3-53/VH3-66 antibodies in immunity against future variants. Recent Omicron variants exhibit active substitution at L452, and we have highlighted the potential role of L452R substitution in evading VH1-69 antibodies[10,36]. YB13-292 showed reduced binding to recent Omicron BA.4/BA.5 spike with L452R, while binding by YB9-258 is not affected by recent Omicron variants (Fig. 2b and Supplementary Fig. 12). Discovery of YB13-292, which mimics VH1-69 antibody L452 recognition mode utilizing the inserted hydrophobic residues in HCDR2, highlights further that the region around L452 as a vulnerability on SARS-CoV-2 spike RBD.

In summary, previous reports observed that prolonged antigen exposure or repeated antigen exposures by SARS-CoV-2 infections or booster vaccines activate somatic hypermutation and affinity maturation. These processes select for more potent and mutation resistant antibodies with neutralization activities not only towards homologous strain but also heterologous variant strains[12,45–48,52]. Our molecular characterization of cross-neutralizing YB9-258 and YB13-292 antibodies identifies that somatic hypermutation likely introduced the identified unusual CDR features which confer cross-neutralizing activities to both YB9-258 and YB13-292. Mutation resistant cross-neutralizing antibodies are likely to play a role in immunity against emerging SARS-CoV-2 variants. The results presented in this study provide further evidence to show importance of somatic hypermutation and affinity maturation in generating mutation resistant cross-neutralizing antibodies.

### Limitation of the study

A major limitation of this study is the use of pooled samples in single-cell sequencing. Therefore, it does not provide characteristics of B-cell response in individual patient and the data would not facilitate more precise comparisons between patients. The antibody selection criteria used may be biased towards identifying cross-reactive antibodies. All synthesized mAbs were primarily tested for binding activity to Delta variant RBD. RBDs of more comprehensive panel of variants could be tested to facilitate better assessment of cross-variant binding mAbs. Furthermore, in vivo protection activities of the cross-neutralizing mAbs were not evaluated.

## Methods

### Data reporting

No statistical methods were used to predetermine sample size. The experiments were not randomized. The investigators were not blinded to allocation during experiments and outcome assessment.

### Study approval

This study and all the relevant experiments were approved by Guangzhou Eighth People's Hospital Ethics Committee (No. 202001134 and 202115202). The research was conducted in strict accordance with the rules and regulations of the Chinese government for the protection of human research participants. We obtained written informed consent from all participants for research use of their blood samples. No compensation is provided for the participants.

### Convalescent patients

A total of 26 COVID-19 patients who had been treated by Guangzhou Eighth People's Hospital, Guangzhou Medical University between Jan 29, 2020 to July 9, 2021 were enrolled in this study (Supplementary Table 1). Of these 26 patients, 11 patients (generating pooled samples: W1#-W2#, W4#-7#, see Supplementary Table 1) were infected by the wildtype (Wuhan) strain in early 2020 without receiving any vaccine. 15 patients (generating pooled samples: YB9#, YB12#-YB14#, see Supplementary Table 1) were infected in the 2021 Guangzhou Delta-variant outbreak and 13 of the 15 patients received one or two doses of inactivated vaccines (Sinopharm BBIBP-CorV or Sinovac CoronaVac) prior to infection. All patients were hospitalized at Guangzhou Eighth

People's Hospital, the designated hospital for treatment of patients with COVID-19 in Guangzhou area. SARS-CoV-2 infection status was verified by RT–PCR of nasopharyngeal swab and throat swab specimens. Blood samples were collected at 2-14 days after patients being tested negative for SARS-CoV-2. Sample collection, processing, and laboratory testing were performed as recommended by China CDC and complied with WHO guidance. These clinical samples are unique biological materials, the participants may or may not consent to give blood samples again, but the same immune responses are unlikely to reproduce.

### PBMCs from blood and antigen-binding B cells sorting

Peripheral blood mononuclear cells (PBMCs) were isolated immediately from fresh blood by Ficoll-Hypaque gradient (GE Healthcare) centrifugation. CD19$^+$ B cells were enriched from pooled PBMCs using a CD19 MicroBeads kit (Miltenyi). The enriched CD19$^+$ B cells were then stained with PE anti-human CD27 antibody (BD Biosciences, Cat# 566944, Clone name: O323, 1:50 dilution), SARS-CoV-2 biotinylated RBD protein (His-tagged) conjugated with FITC-streptavidin (Biolegend, Cat# 405202, 1:20 dilution), and biotinylated S1 protein (His-tagged) conjugated with APC-streptavidin (Biolegend, Cat# 405243, 1:20 dilution). CD19$^+$CD27$^+$RBD$^+$S1$^+$ B cells were sorted with a BD AriaFusion flow cytometer. The purity of sorted cells was rechecked by FACS again on a BD AriaFusion. Sorted CD27$^+$Delta-S1$^+$Delta-RBD$^+$ B cells were resuspended in PBS containing 2% (v/v) fetal bovine serum (FBS) for future use. Flow cytometry data were analyzed using FlowJo v10.

### Single-cell library preparation, sequencing, and alignment

FACS sorted CD27$^+$Delta-S1$^+$Delta-RBD$^+$ memory B cells were loaded onto a 10× Chromium A Chip. Single-cell lysis and RNA first-strand synthesis were performed using 10× Chromium Single Cell 50 Library & Gel Bead Kit according to the manufacturer's protocol. After RNA first-strand synthesis, samples were heated to 85 °C for 5 min to ensure denaturation of possible infectious materials. RNA and V(D)J library preparation was performed according to the manufacturer's protocol (Chromium Single Cell V(D)J Reagent Kits, 10× Genomics). Single-cell transcriptome data and V(D)J data were processed using Cellranger (v.6.1.2) pipeline. Gene expression matrices were generated by alignment to human reference GRCh38. To annotate the V(D)J sequences, custom reference was built by Cellranger mkref pipeline based on the human germline immunoglobulin gene sequences from IMGT (https://www.imgt.org/). The output filtered data was analyzed by R software (v.4.1.2).

### Single-cell transcriptome data processing

Gene expression matrices were analyzed by the Seurat package (v.4.0.5)[15]. A preliminary selection of high-quality single cells which meet the following criteria was performed: (1) > 200 genes and <4000 genes; (2) <10% gene derived from the mitochondrial genome; (3) unique B-cell receptor (BCR). To remove batch effects, the standard Seurat v3 integration workflow was applied to datasets from each sample. Briefly, raw counts of each sample were normalized using the NormalizeData function and 2000 highly variable genes were selected using the FindVariableFeatures function in each sample. Then, 'anchors' between pairs of samples were identified with the FindIntegrationAnchors function. Finally, a batch-corrected expression matrix of all cells was created by the IntegrateData function according to these anchors.

### Cell-type annotation and differential expression analysis

After integration, principal component analysis (PCA) was performed on the scaled gene expression matrix. To cluster the cells, a graph-based clustering approach built upon initial strategies developed by Macosko and colleagues was applied[55]. Briefly, a KNN graph was

constructed based on 30 principal components using the FindNeighbor function. Then, cells were clustered using the FindClusters function with a resolution of 1 and visualized by Uniform Manifold Approximation and Projection (UMAP). The R package Single R (v.1.8.0)[56] was used to annotate single cells with Monaco Immune Cell Data as reference[57]. Cell-type annotation was further modified according to the expression of classic marker genes. Differential gene expression analysis was performed using the FindAllMarkers function with default parameters. Genes were claimed as significantly differentially expressed if: (1) log fold change > 0.25; (2) adjusted P value <0.05.

## Single-cell V(D)J data processing

Single-cell V(D)J data was firstly filtered by retaining cells that express unique BCR with one productive heavy chain (IGH) and one productive light chain (IGK/IGL). Somatic hypermutation of BCR was identified by IgBlast (v.1.18.0)[58] using amino acid sequences. The similarity between BCRs is defined by the following conditions: (1) encoded by the same IGHV and IGKV/IGLV genes; (2) more than 80% identical to the amino acid sequences of HCDR3 or LCDR3 (identity determined by Hamming distance).

## Antibody production

Expression and purification of recombinant antibodies were performed on the GenScript's high-throughput CHO cell manufacturing platform. The recombinant antibody sequences were codon-optimized for Chinese Hamster Ovary (CHO) cells and synthesized by GenScript. The optimized DNA fragment (heavy chain and light chain) was cloned into the Genscript optimized expression vector, before being transfected into CHO cells. Transfected CHO cells were cultured with GenScript's optimized expression process. Antibodies were purified and quality-checked by SDS-PAGE and A280. Purified antibodies (IgG) were used in binding or neutralizing assays.

## Quantification of WT-RBD-specific IgG antibodies

Plasma samples were inactivated at 56 °C for 30 min before testing. IgG antibodies against the SARS-CoV-2 WT-RBD protein were quantified using a two-step indirect immunoassay electrochemiluminescence immunoassay kits (Antu Biotech Co., Ltd.), according to the manufacturer's instructions. Briefly, the samples were first incubated with microparticles coated with the SARS-CoV-2 WT-RBD protein and acridine ester-labeled antibodies against the Fc domain of human antibodies. After washing, quantification of bound IgG was performed on an automatic chemiluminescence immunoanalyzer (AutoLumo A1000, Antu Biotech Co., Ltd). All tests were performed under strict biosafety conditions. Antibody levels are presented as the measured chemiluminescence values divided by the cut-off (cut-off index, COI). COI < 1 was regarded as negative, and COI > 1 was regarded as positive.

## ELISA quantification

ELISA plates were coated with RBD or S protein of SARS-CoV-2 and VOCs (Sino Biological) at 0.5 μg/ml in PBS. The expressed mAbs were detected by indirect ELISA using a SARS-CoV-2 antibody detection kit (Sino Biological). According to the manufacturer's instruction, mAbs were defined as ELISA-positive when the $OD_{450}$ value ≥ 2.1 times the mean absorbance value of negative controls. Data were analyzed using Graphpad Prism 8.0.

## Pseudovirus neutralization assay

Lentivirus based pseudoviruses were produced by transfecting 293T cells (ATCC, CRL-3216) with plasmids. Briefly, plasmids carrying SARS-CoV-2 spike gene and lentivirus backbone were co-transfected into 293T cells, and pseudoviruses were harvested from supernatant. Pseudovirus titers were evaluated by luciferase assays on 293T-hACE2 cells as previously described[59]. Various concentrations of mAbs (3-fold

serial dilution using DMEM) were mixed with the same volume of SARS-CoV-2 pseudovirus with a $TCID_{50}$ of $2 \times 10^4$ in 96-well plate. The mixture was incubated for 1 h at 37 °C and subsequently transferred to 293T-hACE2 cells. After ~72 h incubation, the relative luminescence unit (RLU) were evaluated by luciferase assay according to the manufacturer's instructions. The inhibition was evaluated by fitting a nonlinear four-parameter dose-response curve to the data using GraphPad Prism 8.0.

## Authentic SARS-CoV-2 neutralization assay

The SARS-CoV-2 viruses 2021XG/Vero-E6/186 (Delta) and 2021XG/Vero-E6/5748 (Omicron BA.1) were obtained from nasal swabs of COVID-19 patients in Guangzhou Eighth People's Hospital and isolated by the Guangdong Provincial Center for Disease Control and Prevention. The tested monoclonal antibodies were serially diluted (4-fold) starting at 5 μg/mL. One hundred and twenty-five microliters diluted antibodies were mixed with equal volume of viruses and incubated for 2 h at 37 °C. The mixture was added onto a monolayer of Vero-E6 (ATCC, CRL-1586) cells in a 96-well plate and incubated for 7 days at 37 °C. After 7 days of incubation, the plates were observed by an inverted optical microscope. The highest dilution that protected more than half of cells from cytopathic effect (CPE) was taken as the neutralization titer. All experiments were performed in a Biosafety Level 3 facility of Guangdong Provincial Center for Disease Control and Prevention.

## Sources of protein constructs

Spike and RBD proteins are generated according to spike sequences of the following strains: Wildtype (hCoV-19/Wuhan/WIV04/2019), Alpha (B.1.1.7), Beta (B.1.351), Gamma (P.1), Kappa (B.1.617.1), Delta (B.1.617.2), Lambda (C.37), Mu (B.1.621), Omicron BA.1 (B.1.1.529), BA.2, BA.2.11, BA.2.12.1, BA.2.13, BA.3 and BA.4.

## Protein expression and purification

All spike proteins used in this study are ectodomains. "S-GSAS/6P/Wildtype" (Wildtype S-6P) was expressed as described previously[60]. Briefly, the gene sequence encoding wildtype SARS-CoV-2 (hCoV-19/Wuhan/WIV04/2019 (GISAID accession no. EPI_ISL_402124)) spike residues 14-1211 was cloned into pCDNA3.1(+) vector with an N-terminal μ-phosphatase signal peptide and a C-terminal TEV-cleavage site, a T4 foldon trimerization motif, and a hexahistidine tag. The protein sequence was modified to remove the S1/S2 multibasic cleavage site (PRRAR to PGSAS), six stabilizing proline substitutions at residues 817, 892, 899, 942, 986, and 987 were introduced. Codon-optimized nucleotide sequence coding for the spike protein of SARS-CoV-2 Omicron BA.1 variant was synthesized commercially (GenScript). "S-R/6P/Omicron BA.1" (Omicron BA.1 S-6P) was generated as described above, except for that the protein sequence was modified to remove the S1/S2 multibasic cleavage site in a different way (PRRAR to R). For structural studies, spikes stabilized by hexa-proline (6P) were used. S-R/Wildtype spike was constructed by changing the multibasic S1/S2 cleavage site PRRAR to a single R according to a previous report without proline stabilization[28]. Based on S-R/Wildtype spike a series of spike mutants were generated and used for BLI study. To express the SARS-CoV-2 RBD, residues 332-527 of spike protein with an N-terminal μ-phosphatase signal peptide and a C-terminal hexahistidine tag was cloned into pCDNA3.1(+) vector. Proteins were expressed and purified following the previously established protocols[10]. Expression and purification of the R1-32 Fab was carried out as previously described[10]. All purified proteins were aliquoted, flash-frozen by liquid nitrogen and stored at −80 °C before use.

## Biolayer interferometry

Binding of mAbs to S proteins was performed on an Octet RED96 instrument (FortéBio, USA), following the protocol described

previously[10]. Briefly, antibodies at 11 μg/ml were immobilized onto Protein A biosensors (FortéBio, USA) to a level of 1.6–1.8 nm and dipped into the wells containing S proteins at various concentrations (200–3.125 nM) for 5 min to observe association. Subsequently, the sensors were transferred into the assay buffer (PBS, PH 7.4, 0.02% Tween 20, 1 mg/ml BSA) for 10 minutes to monitor dissociation. To measure binding of mAbs to variant RBDs, sensors immobilized with antibodies were dipped into wells containing RBDs (200 nM) for 5 min, followed by 10 min dissociation. Uncoated biosensors were also dipped into S or RBD solutions and the buffer to record references. Data were reference-subtracted and analyzed using Data Analysis HT Version 12.0.2.59 software (FortéBio) with a 2:1 fitting model for binding to spikes (with calculated kinetic parameters shown in Supplementary Table 2) and 1:1 fitting model for binding to RBDs (with calculated kinetic parameters shown in Supplementary Tables 3, 4 and 5). Raw data and fits were plotted in GraphPad Prism 8.0.

## Cryo-EM sample preparation and data collection

All spikes used for cryo-EM studies are stabilized by the introduced 6 prolines (6P)[60]. For the Omicron BA.1 S-6P:YB9-258 Fab and Omicron BA.1 S-6P:YB13-292 Fab complexes, Omicron BA.1 S-6P at 3.1 mg/ml was mixed separately with YB9-258 Fab and YB13-292 Fab at a 3:3 (S monomer:Fab) molar ratio and incubated for 5 min and 1 min, respectively. Three microliters of the mixture supplemented with 0.1% octyl-glucoside (Sigma-Aldrich, V900365) was applied to freshly glow discharged (at 15 mA for 30 s, GloCube, Quorum) holey carbon grids (Quantifoil, Cu R1.2/R1.3). The grids were plunge-frozen in liquid ethane using a Vitrobot Mark IV (ThermoFisher Scientific) with a blot force of 2 and 4 s blot time at 22 °C and 100% humidity. Cryo-EM data were collected using a Talos Arctica electron microscopy (Thermo-Fisher Scientific) operated at 200 keV and equipped with a K3 direct electron detector (Gatan). Movies were recorded using SerialEM at a nominal magnification of 45,000× with a calibrated pixel size of 0.88 Å and a defocus range between −0.8 and −2.5 μm. Each movie was collected with an exposure time of 1.86 s and a dose rate of 25 e⁻/pixel/s, which resulted in a total dose of 60 e⁻/Å² over 27 frames.

For the Wildtype S-6P:YB9-258 Fab:R1-32 Fab complex, Wildtype S-6P at 4.43 mg/ml was mixed with YB9-258 Fab and R1-32 Fab at a 3:3:3 molar ratio and incubated for 1 min. Cryo-EM grids were prepared as described above and loaded onto a 300 keV Titan Krios electron microscope equipped with a Falcon4 direct electron detector with SelectrisX energy filter (slit width 10 eV). Automated data collection by EPU software (ThermoFisher Scientific) was performed in super-resolution counting mode at a nominal magnification of 165,000× with a super-resolution pixel size of 0.366 Å/pix on the image plane, and with a defocus range between −0.6 and −2.0 μm. Each movie was collected with the electron event representation (EER) mode and recorded with a dose rate of 5.32 e⁻/pixel/s and a total dose of 50 e⁻/Å².

## Cryo-EM data processing

The following data processing procedures were summarized in Supplementary Fig. 4.

For Omicron BA.1 S-6P:YB9-258 Fab dataset, movies were aligned in RELION v4.0[61] using MotionCor2 algorithm[62], CTF-estimation and template-free particle picking were performed in Warp[63]. Picked particles were then imported back into RELION v4.0. An EM structure of SARS-CoV-2 S protein in closed form (EMD-11333, https://www.ebi.ac.uk/emdb/EMD-11333)[28] was filtered to 50 Å resolution as the initial reference in the first and second 3D classifications. First round of 3D classification was performed at 3× binning to identify S protein trimers. S trimers 3D classes displaying good structure features were pooled and subjected to one round of 2D classification to remove bad particles. Subsequently, a second round of 3D classification was used at 1.875× binning to separate spikes into different conformations. In the

Omicron BA.1 S-6P:YB9-258 Fab dataset, 3:1, 3:2 but not 3:3 S:Fab structures were observed in the second round of 3D classification. 3:1 structure was auto-refine to a 6.21 Å final map. 3:2 structures were combined and subjected to iteratively auto-refinement, CTF refinement and Bayesian polishing. Then a further 3D classification was performed for 3:2 dataset with a reference map in locked form (EMD-11331, https://www.ebi.ac.uk/emdb/EMD-11331)[28] filtered to 50 Å to remove more bad particles. A final 3:2 map with overall resolution of 4.31 Å was obtained. In order to improve resolutions for flexible Fab bound RBD and "up" - "down" RBD interaction region (RBD-dimer), particles of the 3:2 structure were subtracted with a focused mask around the NTD-RBD-Fab region (Fab focused) and the NTD-RBD-dimer-Fab VHL region (RBD-dimer focused) respectively. Subtracted particles were subjected to reconstruction with local refinement in cryoSPARC. A RBD-dimer focused map with a resolution of 4.54 Å was yielded after one round local refinement. Fab focused structure was refined with a further no alignment 3D classification with 0.1 class similarity. A final Fab focused map with a resolution of 4.69 Å was obtained after a second round of local refinement.

For the Wildtype S-6P:YB9-258 Fab:R1-32 Fab datasets, within cryoSPARC live, movies were patch motion corrected, dose weighted and CTF estimated. Particles were reference-free picked by blob picker for 2D classification. Well featured particles were selected as templates for template picking in all recorded images. Two rounds of reference-free 2D classification were performed at 3× binning using cryoSPARC to remove bad particles. After second 2D classification, complexes of S and S1 were clearly observed. Particles of S complex and S1 complex were selected for two rounds of ab-initio reconstruction at 2× binning with 0.1 and 0.3 class similarity respectively, in sequence, to generate initial models and further remove bad particles. 3D refinements were carried out using non-uniform refinement with default parameters in cryoSPARC. Final maps of Wildtype S-6P:YB9-258 Fab:R1-32 Fab (3.54 Å overall resolution) and Wildtype S1:YB9-258 Fab:R1-32 Fab (3.43 Å overall resolution) were obtained.

Omicron BA.1 S-6P:YB13-292 Fab dataset was processed with similar method used for processing the Omicron BA.1 S-6P:YB9-258 Fab dataset. All 3:3 structures of Omicron BA.1 S-6P:YB13-292 Fab particles were combined for Fab focused refinement. To avoid losing conformations, we performed 3D classification again using particles from 2D classification with a reference map in locked form (EMD-11331)[28] filtered to 50 Å. Conformations of 0 RBD "up", 1 RBD "up" and 2 RBD "up" map were observed in 3D classification. Particles of these different conformations were selected respectively to be 3D refined. Overall resolutions of Fab focused, 0 RBD "up", 1 RBD "up" and 2 RBD "up" maps were 3.95, 6.03, 4.35, and 4.18 Å, respectively.

Local resolution estimation, filtering, and sharpening were carried out using RELION v4.0. Reported resolutions are based on the gold-standard Fourier shell correlation (FSC) of 0.143 criterion and Fourier shell correlation curves were corrected for the effects of soft masking by high-resolution noise substitution.

## Model building and analysis

A previously determined structure of SARS-CoV-2 Omicron BA.1 spike protein with ACE2 bound (PDB: 7T9K)[45] or wildtype spike protein with ACE2 bound (PDB: 7A98)[64] was fitted into the refined maps and used as the starting models. Structures of YB13-292 Fab H and L chains were generated from PDBs 7FG2 and 5BK5 respectively. Structures of YB9-258 Fab H and L chains were generated from PDBs 7K8M and 4PY7, respectively. These structures were fitted into cryo-EM maps in UCSF Chimera v1.14[65]. Iterative model building and real space refinement were performed in Coot v0.9.6[66] and PHENIX[67]. Coordinates of NAGs were manually placed into the corresponding map densities in Coot. Model refinement statistics are summarized in Supplementary Table 6. Interfaces analysis was performed by PISA[68]. Figures were generated using UCSF chimera v1.14.

**Reporting summary**

Further information on research design is available in the Nature Portfolio Reporting Summary linked to this article.

## Data availability

Single B-cell transcriptome and VDJ sequencing data reported in this study have been deposited at the National Genomics Data Center (https://bigd.big.ac.cn/) under the accession number: PRJCA012020. Cryo-EM density maps for the SARS-CoV-2 Wildtype and Omicron BA.1 spikes (S-6P) in complex with YB9-258 Fab have been deposited in the Electron Microscopy Data Bank (EMDB, https://www.ebi.ac.uk/emdb/) with accession codes EMD-34649, EMD-34650, EMD-34651, EMD-34652, EMD-34653 and EMD-34654. The related atomic models have been deposited in Protein Data Bank (PDB, https://www.rcsb.org/) under accession codes 8HC2, 8HC3, 8HC4, 8HC5, 8HC6 and 8HC7, respectively. Cryo-EM density maps for the SARS-CoV-2 Omicron BA.1 spike (S-6P) in complex with YB13-292 Fab have been deposited in the Electron Microscopy Data Bank (EMDB) with accession codes EMD-34655, EMD-34656, EMD-34657 and EMD-34658. The related atomic models have been deposited in Protein Data Bank (PDB) under accession codes 8HC8, 8HC9, 8HCA and 8HCB. Source data are provided with this paper.

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

## Acknowledgements

We thank the staffs at Cryo-EM Facility of GIBH-CAS and Guangzhou Laboratory Bio-imaging Technology Platform for help on cryo-EM sample preparation and data collection. This work was supported by Emergency Key Program of Guangzhou Laboratory (EKPG21-06 to X.X., EKPG21-29 to X.T. and EKPG21-31 to F.L.); R&D Program of Guangzhou Laboratory (SRPG22-002 to X.X. and SRPG22-003 to J.H.); Key R&D Program of Guangdong Province (No. 2021A1111110002 to H.Y.), National Key Research and Development Program of China (2021YFC0863300 to X.T.), Zhongnanshan Medical Foundation of Guangdong Province (No. ZNSA-2021004 to X.T.), National Natural Science Foundation of China (82041014 to L.C. and 202100192 to X.X.). Natural Science Fund of Guangdong Province (2021A1515011289 to X.X. and 2022A1515110495 to B.L.). X.X. acknowledges Start-up grants from the Chinese Academy of Sciences and Bioland Laboratory (GRMH-GL).

## Author contributions

H.Y., M.W., X.Liu. and X.T. conceived and initiated antibody isolation from patients with funding obtained by X.T. and F.L.; H.Y., Y.H., G.T., J.L. and L.G. organized the collection of convalescent patients' blood samples and defined parameters for FAC sorting. H.Y., L.G., Y.W. and J.L. prepared B cells and performed 10× sequencing. H.Y. and S.Q. performed scRNA and scVDJ library preparation; Y.Z., X.Liu, H.X., J.S., C.L. and Q.Y. analyzed the sequencing data; H.Y. and M.W. selected the antibody sequences for validation; Q.W., P.H., C.F., P.Z. and X.G. performed the pseudovirus neutralization assays; X.Lin and C.X. performed SPR assays; B.L. Q.C. and M.S, performed BLI assays; D.L., B.K., C. K., X.P., R.P. and X.C. performed authentic virus neutralization assays; B.L. and Q.C. purified proteins for cryo-EM samples under the supervision of X.X.; B.L. and X.G. collected cryo-EM data under the supervision of J.H. and X.X.; X.G. and Z.L. processed cryo-EM data under the supervision of J.H. and X.X.; X.G. and X.X. analyzed cryo-EM structures and prepared the

figures; X.X., H.Y., Y.Z., B.L. and M.W. wrote the paper with input from all authors. N.Y., L.L., F.H., F.L., and L.C. supervised the research.

## Competing interests

H.Y., M.W. and X.T. are co-inventors on patent applications describing the neutralizing mAbs (patent application number: CN202210313522.1). The other authors declare no competing interests.

## Additional information

[1]Guangzhou Eighth People's Hospital, Guangzhou Medical University, Guangzhou, China. [2]The State Key Laboratory of Respiratory Disease, CAS Key Laboratory of Regenerative Biology, Guangdong Provincial Key Laboratory of Stem Cell and Regenerative Medicine, Guangdong Provincial Key Laboratory of Biocomputing, Center for Cell Lineage and Development, Guangzhou Institutes of Biomedicine and Health, the Chinese Academy of Sciences, Guangzhou, China. [3]University of Chinese Academy of Sciences, Beijing, China. [4]Bioland Laboratory (Guangzhou Regenerative Medicine and Health—Guangdong Laboratory), Guangzhou, China. [5]State Key Laboratory of Respiratory Disease, National Clinical Research Center for Respiratory Disease, Guangzhou Institute of Respiratory Health, the First Affiliated Hospital of Guangzhou Medical University, Guangzhou, China. [6]BGI-Shenzhen, Shenzhen, China. [7]China National GeneBank, BGI-Shenzhen, Shenzhen, China. [8]Guangdong Provincial Center for Disease Control and Prevention, Guangzhou, China. [9]Guangzhou Laboratory, Guangzhou International Bio Island, Guangzhou, China. [10]University of Science and Technology of China, Hefei, Anhui, China. [11]State Key Laboratory of Virology, Center for Biosafety Mega-Science, Wuhan Institute of Virology, Chinese Academy of Sciences, Wuhan, China. [12]These authors contributed equally: Haisheng Yu, Banghui Liu, Yudi Zhang, Xijie Gao, Qian Wang, Haitao Xiang, Xiaofang Peng, Caixia Xie, Yaping Wang. ✉e-mail: yuhaisheng@gzhmu.edu.cn; kecw1965@aliyun.com; gz8h_lifeng@126.com; he_jun@gibh.ac.cn; wangmeiniang@genomics.cn; chen_ling@gibh.ac.cn; xiong_xiaoli@gibh.ac.cn; tangxp@gzhmu.edu.cn

