## [Peer Review File · Nature Communications]

Somatically hypermutated antibodies isolated from SARS-CoV-2 Delta infected patients cross-neutralize heterologous variantsREVIEWER COMMENTS

Reviewer #1 (Remarks to the Author):

“SARS-CoV-2 breakthrough infections induce somatically hypermutated broadly neutralizing antibodies against heterologous variants”

In this work the authors aimed to understand the molecular aspects of neutralizing antibodies elicited following breakthrough delta infection to evolving SARS-CoV-2 variants of concern (VOC). The experiments utilized pooled CD19+CD27+ S1+(Delta RBD) MBC and 10x Chromium to infer V(D)J sequences. The resulting IgG1 monoclonal antibodies (mAbs) were purified, and characterized for binding and neutralization. Two of the mAbs (YB9-258 and YB13-292) potently neutralized omicron. The authors utilized Cryo-EM to elucidate the molecular characteristics of the antibodies that neutralized Omicron and showed that SHM and CDRH2 insertion mediated expansion of breadth towards elicitation of the omicron neutralizing response. It is encouraging to observe these responses particularly given the vaccination and infection with delta strain of SARS-CoV-2. The manuscripts is well written, the results are explained clearly and the claims of the study are justified by the data.

A few suggestions:

1. Given the extremely low incidence of omicron neutralizing mAbs (two participants in fourteen subjects) in previously vaccinated individuals encountering breakthrough infections. It will be helpful if the authors show the ELISA binding responses of individual sera binding Omicron, across VOCs (provided the sample is present).
2. Also, given these are elicited by previously established public clonotypes (class 1 VH3-53, class 2 VH3-21), it will be of interest to provide insights into the fraction of antibodies of public clonotype that are undergoing such SHM events.
3. It will be of interest to highlight the differences between YB13-281 and YB13-292. Given both use the same VH3-21 while only YB13-281 binds SARS-CoV-1 and Omicron (Figure 1i, 2B).
4. Given the ongoing evolution of SARS-CoV-2. Maybe the use of the term “broadly” neutralizing in the title may not be advisable.
5. The authors should clearly mention the rate constants ($k_{on1/2}$ and $k_{off1/2}$) used for comparing the apparent KD. Provided the authors used 2:1 binding model for determining the apparent KD.
6. The local resolution and map fits needs to be provided at the binding interface. To provide clarity of the model to map.

Reviewer #2 (Remarks to the Author):

The manuscript by Yu et al., describes the isolation and characterization of monoclonal antibodies from vaccinated and unvaccinated patients that were infected with the SARS-CoV-2 Delta variant. The noteworthy results were the structural characterization and modeling of two of the most potent mAb, YB9-258 and YB13-292. YB9-258 presumably showed highly mutated residues while YB13-292 showed an insertion of SNIL motif into the antibody. Whilst these findings are novel and useful to the field, the manuscript is weakened by some over-interpretation of data in the first part of the manuscript.

One major flaw in the first part of the manuscript is that the authors used pooled B-cell samples from 14 vaccinated and 9 unvaccinated samples. As these data were derived from pooled samples, essentially all observations/results in line 110 onwards would merely represent “an average”. The authors should be cautious in overgeneralizing that all patients with breakthrough infections will show increased SHM. Most patients in the unvaccinated pool were infected with the wild-type SARS-CoV-2

strains and all samples were sorted using WT-antigens including the 3 patients infected with Delta/Gamma strains. This could potentially introduce some bias in the study and should be acknowledged as a limitation of the study.

Another concern is that the authors posit that SHM drives increased diversity and frequencies of broadly neutralizing antibodies, leading to enhanced immunities towards VOC and provided as examples, the structural mechanisms of neutralization for the 2 Mab. That repeated exposure increases SHM which increases antibody diversity is not a novel concept, however, only 24/110 (20%) binding Mab cross-reacted with other strains. Without an appropriate baseline/control group, it is hard to say that this represented enhanced immunity. The authors should test the polyclonal sera to demonstrate whether these breakthrough infection really conferred enhanced immunity. Otherwise, these B-cell clones may represent only rare subsets that do not contribute significantly to enhancing cross-reactivity.

Furthermore, while some information on the patients were given in Extended Data Table 1, I do not believe this information was sufficient to fully understand the data. Some missing information:

- i) Line 671 indicated that some patients only received a single dose of vaccine, however this was not indicated in Ex. Table 1 at a patient level. Furthermore, what does \ mean?
- ii) Blood sample was collected 7-14 days after discharge. Would be better if this was specifically indicated for each patient. It would also be useful to indicate the time between last vaccine dose and blood sampling time.
- iii) No vaccination or antibody titers at a polyclonal level for all patients?

Other concerns:

1. SHM is a cellular process that produces the antibodies. It is incorrect to refer to antibodies as SHM-antibodies. E.g Line 73-74 and several places throughout MS.
2. Similarly, it is also incorrect to say Omicron evades neutralization by acute infection induced germline antibodies. Omicron is more resistant to neutralization by antibodies induced by strains circulating during earlier stages of pandemics even at a polyclonal level.
3. Several language issues- meaning of sentences are unclear.
4. Abstract needs to be reworked.

Reviewer #3 (Remarks to the Author):

Yu and colleagues describe in this manuscript the molecular mechanism for enhanced efficacy of affinity matured neutralizing antibodies against heterologous variant of concerns (VOC). The authors analyzed memory B cells (MBC) from Delta breakthrough infections and found that they express antibodies with higher numbers of somatic hypermutations (SHMs). They isolate several antibodies able to bind the Omicron variant BA.1 and structurally analyzed two of them. Their analysis show how the variation introduced by the SMHs contribute to increase the antibody contact area with the epitope. The authors conclude that booster vaccines by increasing the evolution of SHMs in the pool of MBC, provide the immune system with resistant antibodies able to counteract the escape mutations present in the new VOC.

It has been reported in the literature that, with the emergence of VOC and the repeated antigen exposure of the population, the antibodies generated after vaccination or re-infection show affinity maturation and the increase in SHMs results in an increased binding to the spike protein (Muecksch F et al. Immunity 2021 - Moriyama S et al. Immunity 2021 – Sokal et al. Immunity 2021). The emergence of Omicron carrying several mutations in the RBD domain of the spike caused several breakthrough infection in the population of vaccinees. However, several authors have identified in vaccinees and infected subjects, up to 30-50% of cross-neutralizing antibodies able to bind this VOC, which are amplified by a booster vaccine eventually (Sokal et al. Immunity 2022 – Lanz et al. <https://doi.org/10.21203/rs.3.rs-1518378/v1> - Garcia-Beltran et al. Cell 2021 – Cameroni et al. Nature

2021). Some of these antibodies in complex with the spike protein have been also structurally studied by cryo electron microscopy (Chi et al. <https://doi.org/10.1038/s41392-022-00987-z> - Sheward et al. <https://doi.org/10.1101/2022.01.03.474825>).

Structural characterization of cross-neutralizing antibodies is necessary to understand how the immune system is coping with the antigenic drift of the virus and how to design more effective vaccines. In this regard the present study is adding new knowledge to the field but it needs to be placed in the context of the published work and the obtained results need to be discussed in comparison to the mechanisms of neutralization proposed in published work.

In particular some point of the paper would need further clarification:

- 1) Line 96: clarify the specificity of the two numbers of B cells, are these S1+ and RBD+ respectively?
- 2) Fig.1b: are these the RBD+ B cells?
- 3) Fig.1d: put the dashed lines in black; it is barely visible. Please add the mean value to these lines. It seems that there is no difference between the lines in Delta breakthrough and Non-vaccinated but the graph shows statistical significance.
- 4) Fig.1i: please indicate why some of the antibodies are marked in red.
- 5) Please add in the text (not only in the excel table) how many of the 110 selected antibodies are S1- and RBD-specific and how many are only S1-specific.
- 6) Correct the title of Table S1; it says Table S2.
- 7) Line 145: Figure 2b is mentioned before figure 2a, please invert the numbering.
- 8) Figure 2, BLI experiment: what is S-R? Please explain. Add "wildtype" to the first row of BLI data.
- 9) Figure 2, neutralization graphs: please indicate that it is pseudovirus.
- 10) Line 146: discrepancy between binding and neutralization has been reported in other studies too.
- 11) Line 152: use "good" instead of "very good" especially in comparison to the binding of YB9-120 and YB13-208.
- 12) Line 156: There is a difference in the resistance to the RBD mutations between the two antibodies with YB9-258 showing resistance to all the mutations tested (Fig.2c and ED Fig. 2a and b). This difference is not highlighted in the text.
- 13) Line 168: additional work can be cited here (Robbiani et al. Nature 2020 – Barnes et al. nature 2020).
- 14) Lines 173-182: please cite work reporting similar results listed in this paragraph such as: Hong et al. Nature 604, 2022 – Guo et al. Cell Reports 39, 2022 – McCallum et al. Science 375, 2022
- 15) Line 187: please explain the rationale for using the complex with two antibodies and cite previous work which used the same strategy (i.e. McCallum et al. Science 2022)
- 16) The binding affinity of YB9-258 and YB13-292 for the RBD of different VOC is different yet both antibodies neutralize Omicron BA.1. In light of the structural data the authors could discuss the molecular basis of this difference. Only the possible mechanism of neutralization is mentioned but not the possible interference with the binding to the different RBDs.
- 17) Lines 293-295: Is it possible that the high number of binding antibodies isolated by this study is the result of a bias introduced by the selection criteria adopted to identify the 270 antibodies studied? The other studies were looking at an unbiased antibody repertoire.
- 18) Line 321: please cite Manner et al. 2022 here and not only in Material and Methods
- 19) Line 553: The yellow masks mentioned in the figure legend are not shown in the ED fig.3

Reviewer #1 (Remarks to the Author):

“SARS-CoV-2 breakthrough infections induce somatically hypermutated cross-neutralizing antibodies against heterologous variants”

In this work the authors aimed to understand the molecular aspects of neutralizing antibodies elicited following breakthrough delta infection to evolving SARS-CoV-2 variants of concern (VOC). The experiments utilized pooled CD19+CD27+ S1+(Delta RBD) MBC and 10x Chromium to infer V(D)J sequences. The resulting IgG1 monoclonal antibodies (mAbs) were purified, and characterized for binding and neutralization. Two of the mAbs (YB9-258 and YB13-292) potently neutralized omicron. The authors utilized Cryo-EM to elucidate the molecular characteristics of the antibodies that neutralized Omicron and showed that SHM and CDRH2 insertion mediated expansion of breadth towards elicitation of the omicron neutralizing response. It is encouraging to observe these responses particularly given the vaccination and infection with delta strain of SARS-CoV-2. The manuscripts are well written, the results are explained clearly and the claims of the study are justified by the data.

We thank the reviewer for his/her favorable comments and constructive suggestions to allow improvement of our manuscript.

A few suggestions:

1. Given the extremely low incidence of omicron neutralizing mAbs (two participants in fourteen subjects) in previously vaccinated individuals encountering breakthrough infections. It will be helpful if the authors show the ELISA binding responses of individual sera binding Omicron, across VOCs (provided the sample is present).

Answer:

Follow the reviewer suggestion, we tested binding of WT, Delta (B.1.617.2) and Omicron BA.1 (B.1.529) RBDs by SARS-CoV-2 convalescent plasmas from patients participated in this study using ELISA (we only included patients after removal of several patient groups due to unconfirmed immunization status, please also see answers to reviewer 2's questions), we included this data in **Supplementary Table 1** and visualized the data in **Supplementary Fig. 1b**. We divided the plasmas into a non-vaccinated WT infected group and a group with Delta-variant breakthrough infections (details for each patient in the two plasma groups are shown in **Extended Data Table 1**, binding data for the 2 patients without vaccination in pooled samples YB9 and YB12 are shown but not included in statistics analysis, so the group is an authentic breakthrough group). The trend of decreasing binding towards succeeding mutant RBDs is obvious within the groups. Although there is some evidence to show that the breakthrough infected group has higher binding ($*p<0.05$) towards Omicron RBD than the non-vaccinated

group, for WT RBD and Delta RBD, we did not observe any significant differences in plasma binding. The immunity enhancement may be weak for the breakthrough group as some of the participated patient only received one dose of vaccine prior to infection. As a result, we feel it is bit overstretching to claim "...enhanced immunity" based on our plasma binding data and we withdrew this conclusion.

2. Also, given these are elicited by previously established public clonotypes (class 1 VH3-53, class 2 VH3-21), it will be of interest to provide insights into the fraction of antibodies of public clonotype that are undergoing such SHM events.

Answer:

We have presented the average putative SHM numbers in VH genes for mAbs of known public clonotype we isolated from our patients (**Fig. 1g**), we found that on average they generally have more putative mutations introduced by SHM. We calculated the percentage of antibodies exhibiting above average SHM in each clonotypes and this is shown in the updated **Extended Data Fig. 1c**. Most VH gene families have a higher proportion of mAbs with more than 5 putative mutations, including VH3-53. However, we didn't see such trend for VH3-21. We added comments "*Comparisons between the binding mAb sequences and known mAb sequences revealed that the accumulation of somatic hypermutations in isolated binding mAbs (Fig. 1g). This is consistent to our observation that B cells from patients primarily with Delta variant breakthrough infections show sign of increased level of somatic hypermutation by single cell V(D)J transcript data (Fig. 1d, g and Extended Data Fig. 1c, d).*" (Line 272-276 in the new tracked-change MS)

3. It will be of interest to highlight the differences between YB13-281 and YB13-292. Given both use the same VH3-21 while only YB13-281 binds SARS-CoV-1 and Omicron (Figure 1i, 2B).

Answer:

We thank the reviewer to point out this interesting result. We have not yet determined the structure of YB13-281, we speculate that the binding mode of YB13-281 is different from that of YB13-292. From the structural study of YB13-292, we found that the HCDR2 insertion is important in epitope binding and likely a major contributor to the specific epitope binding mode. Investigation of several structurally characterised VH3-21-encoded mAbs revealed that they can target multiple epitopes, including previously defined class 1 (ADG20 (PDB:7U2D)), class 2 (BD-804 (PDB:7EYA) and YB13-292) epitopes on RBD. Several VH3-21 mAbs bind the neutralizing supersite on NTD (S2L28 (PDB: 7LXX), N9 (PDB:7E8F), P008_056 (PDB:7NTC)), therefore, it is difficult to pin down exactly where YB13-281 would bind without structural study.

Nevertheless, we highlighted differences between YB13-281 and YB13-292 in the Discussion section with an extra sentence (line 599-603 in the new tracked-change MS): "In addition to YB13-292, there is another VH3-21-encoded mAb isolated from this study, YB13-281, which

displays cross-reactivity not only towards RBDs of SARS-CoV-2 variants but also SARS-CoV-1 (Fig. 1i). Different from YB13-292, YB13-281 utilizes VL1-40 for light chain and this might be responsible for its unique activity.”

4. Given the ongoing evolution of SARS-CoV-2. Maybe the use of the term “broadly” neutralizing in the title may not be advisable.

Answer:

We thank the reviewer for this suggestion and we have replaced “broadly neutralizing” with “**cross-neutralizing**” in the title. In addition, we have changed “broadly neutralizing” to “**cross-neutralizing**” at 4 places throughout the original text. We have also refrained from using “broadly neutralizing” in newly added text.

5. The authors should clearly mention the rate constants ($k_{on1/2}$ and $k_{off1/2}$) used for comparing the apparent K_D . Provided the authors used 2:1 binding model for determining the apparent K_D .

Answer:

We apologize for this confusion. To briefly explain, in this study, all binding experiments were done using Protein A biosensors to immobilize IgG. We used either monomeric his-tagged RBDs or trimeric spikes as analytes to measure the interactions. Because the RBDs are monomeric, the binding sites are independent, the 1:1 binding model was able to describe the binding curves, and for all the binding experiments with RBDs, only one k_{on} , k_{off} and K_D were reported (these are summarized in **Extended Data Tables 3, 4 and 6**). For binding experiments with spikes, likely due to multivalent binding, we found 2:1 model was required to fit the binding curves properly, hence we used the 2:1 binding model and reported k_{on1} , k_{on2} , k_{off1} , k_{off2} and associated K_{DS} (reported in **Extended Data Table 2**).

For comparison of K_D values, we only used RBD binding data, so only one K_D from each binding curve (derived from k_{off}/k_{on}) was involved in comparison. **Fig. 2c** used RBD binding data shown in **Extended Data Fig. 2a, b** with $k_{on}/k_{off}/K_D$ values summarized in **Extended Data Table 3**. For **Extended Data Fig. 10d** used RBD binding data shown in **Extended Data Figs. 2b and 10a-c** with $k_{on}/k_{off}/K_D$ values summarized in **Extended Data Tables 3 and 4**. And for **Extended Data Fig. 11 (lower panel)** used RBD binding data shown in **Extended Data Fig. 11 (upper panel)** with $k_{on}/k_{off}/K_D$ values summarized in **Extended Data Table 6**. We thank the reviewer for pointing out this unclarity, and in Figure legends of **Fig. 2c and Extended Data Figs. 10d and 11**, we added sources of K_D data used for comparison.

We have also included more details “Data were reference-subtracted and analyzed using Data Analysis HT Version 12.0 software (Fortebio) with a 2:1 fitting model for binding to spikes (with calculated kinetic parameters shown in **Extended Data Table 2**) and 1:1 fitting model for binding to RBDs (with calculated kinetic parameters shown in **Extended Data Tables 3, 4 and 6**)” in the updated Methods section (line 1413-1415 of the new tracked-change MS) regarding to binding models used for spike and RBD data respectively to clarify the data analysis

procedure.

6. The local resolution and map fits needs to be provided at the binding interface. To provide clarity of the model to map.

Answer:

We thank the reviewer for the suggestion. We updated **Extended Data Fig. 4d** and **Extended Data Fig. 4i** to include local resolution assessments zoomed in at the antibody-antigen interfaces.

Reviewer #2 (Remarks to the Author):

The manuscript by Yu et al., describes the isolation and characterization of monoclonal antibodies from vaccinated and unvaccinated patients that were infected with the SARS-CoV-2 Delta variant. The noteworthy results were the structural characterization and modeling of two of the most potent mAB, YB9-258 and YB13-292. YB9-258 presumably showed highly mutated residues while YB13-292 showed an insertion of SNIL motif into the antibody. Whilst these findings are novel and useful to the field, the manuscript is weakened by some over-interpretation of data in the first part of the manuscript.

Answer:

We thank the reviewer to point out the novelty and limitations of the study. We appreciate the constructive suggestions offered for manuscript improvement. We made detailed modifications after studying the reviewer's comments.

One major flaw in the first part of the manuscript is that the authors used pooled B-cell samples from 14 vaccinated and 9 unvaccinated samples. As these data were derived from pooled samples, essentially all observations/results in line 110 onwards would merely represent "an average". The authors should be cautious in overgeneralizing that all patients with breakthrough infections will show increased SHM. Most patients in the unvaccinated pool were infected with the wild-type SARS-CoV-2 strains and all samples were sorted using WT-antigens including the 3 patients infected with Delta/Gamma strains. This could potentially introduce some bias in the study and should be acknowledged as a limitation of the study.

We thank the reviewer to point out limitations in this study. We would like to explain the use of pooled sample first. As the designated hospital for COVID-19 treatment in the Guangzhou area, we are responsible exclusively for treatment of all COVID-19 patients from cities of Guangzhou, Foshan and Dongguan. Based on understanding of helping better understanding of COVID-19, some patients kindly agreed to donate blood for research, however, they would normally decline further sample collection after quarantine (they were quarantined after hospital discharge) as it would be too much of a bother for their daily lives. Therefore, although we could recruit many patients, their blood samples are very precious. In addition, our primary aim was to isolate neutralising antibodies. Our preliminary experiment showed that under the setup we have, we encountered low cell sorting efficiency for certain samples. In order to increase the number of recovered cells for single cell sequencing and to maximise SARS-CoV-2 specific B cell recovery, we pooled the samples. Lastly, according to previous experience in our laboratory, single-cell transcriptome sequencing with fresh blood has been more effective, we had to pool blood samples collected in the same day on which the patients agreed to donate blood for the last time before their quarantine ended to allow samples to be processed on the same day. By

analysing single cell sequencing data derived from the samples prepared this way, we were able to isolate a few cross-neutralising antibodies. We do apologise for confusions may be caused by sample mixing and in the updated MS we attempt to describe the overall characteristics of the single cell sequencing data while try to minimize over-interpretation of the data following the suggestions offered by the reviewer.

After studying the reviewer's careful review report, we found inaccuracy in the patient sample information, therefore we want to address reviewer's **point i)** first.

Furthermore, while some information on the patients were given in Extended Data Table 1, I do not believe this information was sufficient to fully understand the data. Some missing information:

i) Line 671 indicated that some patients only received a single dose of vaccine, however this was not indicated in Ex. Table 1 at a patient level. Furthermore, what does \ mean?

Answer:

We thank the reviewer's careful review of the manuscript. In the original table, the "\ " means "vaccine status not confirmed". We apologise for including these samples and confusion caused. Following the reviewer's suggestion, we carefully reviewed patient information, and we found several patients with immunization status that needs further confirmation, we removed groups with such patients (but sequencing data derived from the removed groups were deposited together with the retained groups: <https://ngdc.cncb.ac.cn/gsa-human/s/ASvWkms>). As a result, we only retained groups YB9, YB12, YB13 and YB14. We also removed groups in our non-vaccinated group who were infected with Delta and Gamma variants following the reviewer's suggestion. The new table now contains more information of individual patient in each group including gender, age, infected virus strain, disease severity, symptoms, vaccine type, vaccine dose, duration of hospitalization, sample collection time, and individual IgG titres for easier understanding of the data. As a result of the above stated change, we reanalysed the data, we updated **Fig. 1 and Extended Data Fig. 1** with the new analysis. Within the retained groups, two patients unfortunately confirmed that they were not vaccinated, so we changed descriptions of the group in several places as "group primarily with Delta-breakthrough infections" or "Delta-infected group" instead to be accurate. We present a table below to summarise the changes to the analysis results.

Figure no.	Revised data	Old MS	New MS		
Figure 1	a	Number of pooled samples for Delta-infected group	14	4	
		Number of cells for single-cell transcriptome data (for Fig. 1b,c,h Delta-infected)	8612	3286	
		Number of cells for single-cell V(D)J data (for Fig. 1d,e,f,g Delta-infected)	9171	3554	
	b	Number of clusters	10	9	
	c	Proportion of memory B cells	88.78%	90.20%	
	d		Number of pooled samples for Non-vaccinated group	9	6
			Number of cells for single-cell V(D)J data (for Fig. 1d,e Non-vaccinated)	6479	4215
			Proportion of B cells with high SHM in Delta-infected group	68.06%	70.19±2.32% (on average)
			Proportion of B cells with unmutated VH sequences in Delta-infected group	not analysed	6.03±0.74% (on average)
			Proportion of B cells with high SHM in non-vaccinated group	59.72%	67.01±1.68% (on average)
			Proportion of B cells with unmutated VH sequences in non-vaccinated group	not analysed	10.73±1.26% (on average)
			Proportion of known mAbs with high SHM	40.01%	40.01%
			Proportion of known mAbs with unmutated VH sequences	not analysed	10.42%
	e	Increased percentage of isotype-switched B cells	10.89%	12.00% (on average)	
	f		Number of synthesized mAbs	270	117
			Number of binding mAbs	110	63
		Number of V gene families for synthetic mAbs	29	19	
h,i	Number of crossing-variant binding mAbs	24	22		

The retained group contributed 39% of the original VDJ sequences, 117/270 of synthesized antibodies and 63/110 binding antibodies, 22/24 cross-reactive antibodies and 6/6 potent cross-reactive antibodies.

According to the new analysis result we revised **Fig. 1 legend**:

(Line 448-470 of the OLD MS) Fig. 1. Single-cell atlas of memory B cells and identification of broadly neutralizing antibodies from breakthrough patients of Delta variant. a, Overview of experimental design. CD27+ Delta-S1+ Delta-RBD+ B cells from breakthrough patients of Delta variant were sorted by fluorescence-activated cell sorting (FACS) and subject to single-cell immune transcriptome sequencing. b, UMAP projection of 8,612 single B cells, showing formation of clusters. Each dot corresponds to a single cell, colored according to identified clusters.d, Density plot showing SHM counts of antibodies from breakthrough-infection patients, non-vaccinated patients and CoV-AbDab (Known mAb). SHM count was defined by numbers of mismatched amino acids on variable heavy-chain gene (VH) using IgBlast. The mean SHM count for each group was indicated by dashed lines. P values were calculated by Wilcoxon test; ***, p < 0.001. The ratio of high/low SHM antibodies is shown in the right panel. High and low are defined by above or below the mean SHM count of known mAbs (in CoV-AbDab16) (mean SHM = 5). e, Fan charts comparing percentage of antibody isotypes between Delta variant breakthrough-infection patients and non-vaccinated patients.

(Revised) Fig. 1. Single-cell atlas of memory B cells and identification of cross-neutralizing antibodies from patients primarily with Delta breakthrough-infection. a, Overview of experimental design. CD27+ Delta-S1+ Delta-RBD+ B cells from patients primarily with Delta breakthrough-infection were sorted by fluorescence-activated cell sorting (FACS) and subject to single-cell immune transcriptome and V(D)J sequencing. Single-cell transcriptome data (3286 cells) are graphically represented in panels b, c and h. Single-cell V(D)J data (3554 cells) are used for the analyses shown in panels d, e, f, and g. b, UMAP projection of B cells shows formation of 9 clusters. Each dot corresponds to a single cell, colored according to identified

clusters.d, Density plot showing SHM counts on variable heavy-chain gene (VH) sequences of B cells from Delta-infected patients, non-vaccinated patients and antibodies from CoV-AbDab (Raybould et al, Bioinformatics, 2021) (Known mAb). SHM count was defined by numbers of mismatched amino acids on VH using IgBlast. The mean SHM count for each group was indicated by dashed lines. The ratio of B cells with unmutated VH sequences is shown in the right panel. P value was calculated by Student's t-test; *, $p < 0.05$. e, Fan charts comparing percentage of 8 antibody isotypes between patients primarily with Delta breakthrough-infection and non-vaccinated patients. Isotypes with significantly higher or lower percentages in Delta-infected group are colored in red or blue, a result from **Extended Data Fig. 1b**.

We agree with the reviewer that previously we simply compared the B cells from each of the two groups, and the results did represent “an average” and may not reflect the true characteristics of each patient. Although we cannot perform analysis on individual patient because we used pooled samples. We attempted to provide extra insights by calculating percentage of B cells expressing unmutated VH genes and percentage of isotype-switched B cells in a per pooled-sample basis in the updated MS. Each pooled-sample is present as data points in the new **Fig. 1d and Extended Data Fig. 1b**. We also removed the non-vaccinated groups infected with Delta/Gamma strains to minimize potential bias.

After removal of the non-vaccinated groups with Delta/Gamma strain infections, we found that there was no significant difference on the average mutation numbers of VH genes expressed by B cells between the two patient groups (shown in the updated **Fig. 1d**). However, we did observe a higher proportion of B cells with unmutated VH genes in non-vaccinated group (added as a new panel in the updated **Fig. 1d**).

Also based on this analysis, the following changes were made:

(Line 110-113 of the OLD MS): “we observed higher proportion of high-SHM ($SHM \geq 5$) antibodies in breakthrough-infection patient samples (68.06%) than in non-vaccinated patient samples (59.72%) and CoV-AbDab (40.01%) which is a collection of 3,984 reported SARS-CoV-2-specific monoclonal antibodies (known mAbs)¹⁶ (Fig. 1d)”.

(Revised) *“on average, we observed lower proportion of B cells expressing unmutated VH genes in patients primarily with breakthrough-infection ($6.03 \pm 0.74\%$) than in non-vaccinated patients ($10.73 \pm 1.26\%$) (Fig. 1d). Recent studies have reported recall of memory B cells in breakthrough infection rather than the activation of naive B cells during primary SARS-CoV-2 infection reported in earlier.”*

(Line 115-117 of the OLD MS): A 10.89% increase in the percentage of isotype-switched antibodies was observed in breakthrough-infection patients, especially a 1.9-fold increase in the proportion of IGHG1 (Fig. 1e). (These are from pooled samples- not sure if you can generalize... should qualify that it represents an average...)

(Revised) *“A 12% increase in the average percentage of isotype-switched B cells was*

observed in patients primarily with breakthrough-infections and for which significantly higher proportion of IGHG1 expression was observed”.

We also updated relevant sections in **Fig. 1 legend for panels d and e**. “**d**, The ratio of B cells with unmutated VH sequences is shown in the right panel. P value was calculated by Student’s t-test; *, $p < 0.05$. **e**, Fan charts comparing percentage of 8 antibody isotypes between Delta-infected patients and non-vaccinated patients. Isotypes with significantly higher or lower percentages in Delta-infected group are colored in red or blue, a result from **Extended Data Fig. 1b**.” (Line 844-849 in the new tracked-change MS).

We updated **Extended Data Fig. 1** with 3 extra panels (panels b, c, d) and relevant info in the legend: (**revised**) “**b**, Proportion of 8 isotypes in each pooled sample, related to **Fig. 1e**. P values were calculated by Student’s t-test; *, $p < 0.05$. **c**, Proportion of mAbs with more than 5 SHM on VH sequences (mean value of known mAbs) for each V gene family, compared between Delta-binding mAbs and known mAbs. **d**, Violin plot showing SHM counts of 22 crossing-variant binding antibodies, 63 binding antibodies and known mAbs. P values were calculated by Wilcoxon test; *, $p < 0.05$, ***, $p < 0.001$.” (Line 1022-1028 in the new tracked-change MS).

We clearly state in the text that the analysis results represent “on average” for the population we isolated the antibodies from.

We also added a “Limitation of the study” section. (Line 656-662 in the new tracked-change MS).

“A major limitation of this study is the use of pooled samples in single-cell sequencing. Therefore, it does not provide characteristics of B cell response in individual patient and the data would not facilitate more precise comparisons between patients. In addition, all synthesized mAbs were primarily tested for binding activity to Delta variant RBD. RBDs of more comprehensive panel of variants could be tested to facilitate better assessment of cross-variant binding mAbs. Furthermore, in vivo protection activities of the cross-neutralizing mAbs were not evaluated.”

Another concern is that the authors posit that SHM drives increased diversity and frequencies of broadly neutralizing antibodies, leading to enhanced immunities towards VOC and provided as examples, the structural mechanisms of neutralization for the 2 Mab. That repeated exposure increases SHM which increases antibody diversity is not a novel concept, however, only 24/110 (20%) binding Mab cross-reacted with other strains. Without an appropriate baseline/control group, it is hard to say that this represented enhanced immunity. The authors should test the polyclonal sera to demonstrate whether this breakthrough infection really conferred enhanced immunity. Otherwise, these B-cell clones may represent only rare subsets that do not

contribute significantly to enhancing cross-reactivity.

Answer:

Follow the reviewer suggestions, we tested binding of WT, Delta (B.1.617.2) and Omicron BA.1 (B.1.529) RBDs by SARS-CoV-2 convalescent plasmas from patients using ELISA (we only included the retained patients after removal of several patient groups), we included this data in **Supplementary Table 1** and visualized the data in **Supplementary Fig. 1b**. We divided the plasmas into non-vaccinated infected groups and a group with Delta-variant breakthrough infections (details for each patient in pooled groups are shown in updated **Extended Data Table 1**, binding data for the 2 patients without vaccination in pooled sample groups YB9 and YB12 are shown within the breakthrough group but not included in statistics analysis, therefore the group is an authentic breakthrough group). The trend of decreasing binding towards succeeding mutant RBDs is obvious within the groups. Although there is some evidence to show that the breakthrough infected group has higher binding ($*p<0.05$) towards Omicron RBD than the non-vaccinated group, for WT RBD and Delta RBD, we did not observe any significant differences in plasma binding. The immunity enhancement may be weak for the breakthrough group as some of the participated patient only received one dose of vaccine prior to infection. Therefore, we agree with the reviewer it is over-stretching to claim “enhanced immunity”. We changed “*These data provide molecular mechanisms for enhanced immunity to heterologous SARS-CoV-2 variants after repeated antigen exposures with implications for future vaccination strategy*” into “*These data provide molecular mechanisms for cross-neutralization of heterologous SARS-CoV-2 variants by antibodies isolated from Delta variant infected patients with implications for future vaccination strategy*” in **line 49-51** of the new abstract.

ii) Blood sample was collected 7-14 days after discharge. Would be better if this was specifically indicated for each patient. It would also be useful to indicate the time between last vaccine dose and blood sampling time.

Answer:

We thank the reviewer for this suggestion. We have included “Sample collection time” for each patient in **Extended Data Table 1**. Blood samples were collected 2-14 days after discharge. We apologize that we cannot provide information about the time interval between the last vaccine dose and the blood sample collection time as we were not able to ascertain the exact time patients were vaccinated. But according to the vaccine roll out schedule in China, and the Delta variant outbreak time in Guangzhou (2021-June) most of the patients should be infected by the Delta variant within 3-6 months of vaccination.

We would also like to clarify the sample collection time, people with COVID-19 were sent to Guangzhou 8th people’s hospital, the designated hospital for COVID-19 treatment in Guangzhou area. Once the patient was tested negative for virus RNA, he or she was discharged from the hospital treatment (this is the discharge time referred in the **Extended Data Table 1**) but was further quarantined for 14 days. Convalescent blood samples were

collected during this quarantine period.

iii) No vaccination or antibody titers at a polyclonal level for all patients?

Answer:

We thank the reviewer for this suggestion. We now included vaccination information and antibody titer for each patient in the updated **Extended Data Table 1**, we also visualized the antibody titer data at a polyclonal level in **Supplementary Fig. 1a**. The antibody titer for all samples were tested for SARS-CoV-2 (WT) specific antibody using a commercially available kit. We included the method for antibody titer detection in the Methods section. (Line 1342-1352 in the new tracked-change MS)

Other concerns:

1. SHM is a cellular process that produces the antibodies. It is incorrect to refer to antibodies as SHM-antibodies. E.g., Line 73-74 and several places throughout MS.

Answer:

We thank the reviewer for this suggestion, we updated almost all sentences about “SHM” in our manuscript and we regard SHM as a cellular process. These modifications include:

Sentence in the abstract: *“These novel features are putatively introduced by somatic hypermutation and they are heavily involved in epitope recognition to broaden neutralization breadth.”*

Line 73-74 of original manuscript: replacing the sentence containing *“generally expressed antibodies with higher numbers of somatic hypermutations”* with *“SARS-CoV-2 antigen specific memory B cells isolated from patients primarily with breakthrough infections of Delta variant are shown to exhibit higher level of somatic hypermutation (SHM)”*. (Line 101-103 in the new tracked-change MS)

Line 120-121 of new MS: *“These two antibodies feature residues introduced by somatic hypermutation”*.

Line 105-106 of original manuscript: replacing *“germline like antibodies with low SHM rates”* with *“germline like antibodies with lower levels of SHM”*. (Line 169-170 in the new tracked-change MS)

Line 117-119 of original manuscript: replacing *“somatic hypermutated mAbs and class switched mAbs”* with *“somatic hypermutated B cells and isotype-switched B cells”*. (Line 183-184 in the new tracked-change MS)

Line 133-135 of original manuscript: replacing *“revealed a trend towards higher SHM for the*

former” with *“revealed the accumulation of somatic hypermutations in isolated binding mAbs”*.
(Line 272-273 in the new tracked-change MS)

Line 383-384 of new MS: *“Residues introduced by SHM are heavily involved in epitope recognition”*

Line 450-451 of new MS: *“selected for antigen binding enhancing changes introduced by SHM”*

Line 453 of new MS: *“due to residues introduced by SHM in YB9-258 LCDR3”*

Line 466 of new MS: *“the P95L change introduced by SHM in YB9-258 LCDR3”*

Line 481-482 of new MS: *“affinity maturation likely selected SHM introduced changes”*

Line 497-498 of new MS: *“The YB13-292 binding interface is dominated by residues putatively undergone SHM”*

Line 557-558 of new MS: *“that residues introduced by SHM facilitate more extensive interactions”*

Line 570 of new MS: *“point changes introduced by SHM”*

Line 573 of new MS: *“In addition to SHM introduced amino acid changes”*

Line 583-584 of new MS: *“like point changes introduced by SHM”*

Line 605-606 of new MS: *“amino acids substitutions and insertions introduced by SHM”*

Line 609 of new MS: *“binding of antibodies showing higher levels of SHM”*

Line 976-977 of new MS (**Fig. 3** legend): *“involving point changes introduced by somatic hypermutation”*

2. Similarly, it is also incorrect to say Omicron evades neutralization by acute infection induced germline antibodies. Omicron is more resistant to neutralization by antibodies induced by strains circulating during earlier stages of pandemics even at a polyclonal level.

Answer:

We thank the reviewer for this suggestion, we replaced the sentence as *“with extraordinary abilities in evading antibodies isolated earlier in the pandemic”* deleting words *“acute-infection induced germline”* in the Abstract.

3. Several language issues- meaning of sentences are unclear.

Answer:

We updated the manuscript according to the PDF kindly provided by the reviewer. These corrections are listed in the last part of this section.

4. Abstract needs to be reworked.

Answer:

We updated the abstract according to suggestions from the reviewers.

Revisions according to the annotated PDF file kindly provided by the reviewer:

Line 1-52 of the new tracked-change MS: We updated Title and Abstract.

Line 63 of the old MS: we spelled out RBD and changed sentence there to "in the receptor binding domain (RBD) alone". (Line 92-93 of the new tracked-change MS)

Line 69 of the old MS: we added "the". (Line 98 of the new tracked-change MS)

Line 73 of the old MS: we corrected throughout the text regarding SHM as a process. These corrections are summarised in responses to "Other concerns point 1".

Line 74-75 of the old MS: We removed "*broadly neutralizing antibodies (bnAbs)*" and changed the sentence to "*We identified a number of cross-reactive neutralizing antibodies (nAbs), with neutralizing activities towards wildtype (WT), Beta and Delta strains.*" (Line 103-104 of the new tracked-change MS)

Line 80 of the old MS: We added the full name of HCDR2: "*heavy chain complementarity determining region 2 (HCDR2)*". (Line 121-122 of the new tracked-change MS)

Line 84 of the old MS: "What specifically is the insight?"

We modified the sentence "*These data give insight into immune responses after repeated exposures of SARS-CoV-2 antigens, ...*" into "*These data provide molecular mechanisms for cross-neutralization of SARS-CoV-2 variants by YB9-258 and YB13-292 isolated from Delta-infected patients*". (Line 125-127 of the new tracked-change MS)

Line 90 of the old MS: "Why pooled and what time point? To check.."

Answer:

PBMCs of COVID-19 convalescent patients were collected 2-14 days after discharge but still under quarantine. We explained the use of pooled samples in the beginning of the point-by-point responses to reviewer 2.

Line 93-94 of the old MS: "how was antigen specific B-cell identified?"

Answer:

CD19⁺ B cells were enriched from pooled PBMCs with Miltenyi CD19 MicroBeads kit. The

enriched CD19⁺ B cells were then stained with PE anti-human CD27 antibody, SARS-CoV-2 biotinylated RBD protein (His Tag) conjugated with FITC-streptavidin, and biotinylated S1 protein (His Tag) conjugated with APC-streptavidin. CD19⁺CD27⁺RBD⁺S1⁺ B cells were sorted with a BD AriaFusion flow cytometer. The purity of sorted cells was rechecked by FACS again with BD AriaFusion, we did not verify the antigen specificity of the sorted B cells with other methods. We added more details in the Methods section. (Line 1237-1269 of the new tracked-change MS)

Line 97 of the old MS: we deleted “*derived from breakthrough-infection patients*” in MS. (Line 140-161 of the new tracked-change MS)

Line 99 of the old MS: We corrected the spelling of “Seurat”. (Line 163 of the new tracked-change MS)

Line 99-100 of the old MS: “What classification was used?”

Answer:

We performed cell clustering following the standard workflow of Seurat. Seurat includes a graph-based clustering approach to cluster cells based on the strategies initial reported in (Macosko et al, Cell, 2015). It is widely used for single-cell transcriptome analysis to distinguish cell subsets based their transcriptional signatures (Melms et al, Nature, 2021; Wilk et al, Nature medicine, 2020; Brioschi et al, Science, 2021; Zhu et al, Immunity, 2020). The number of clusters in **Fig.1b** was determined by the resolution parameter of FindClusters function, with increased values leading to a greater number of clusters. It is reported in the Seurat guided tutorial that resolution between 0.4-1.2 typically returns good results for single-cell datasets of around 3000 cells (https://satijalab.org/seurat/articles/pbmc3k_tutorial.html). Therefore, we set a resolution of 1 to ensure that more clusters were identified. After clustering, we annotated the cells using SingleR, an automatic annotation method for single-cell RNA sequencing data. It is also used in many studies (Cao et al, Cell, 2020; Huuhtanen et al, Nat Commun, 2022; Mair et al, Nature, 2022). Labels of cells were determined by their similarity to the reference dataset with known labels. We also manually modified the labels according to the expression of classic marker genes such as IGHC genes.

We added more details in the “Cell-type annotation and differential expression analysis” part in the Methods section:

(Line 711-714 of original manuscript): After integration, principal component analysis (PCA) was performed on the scaled gene expression matrix. 30 principal components were used to construct a KNN graph. Cells were clustered using the FindClusters function with a resolution of 1 and visualized by Uniform Manifold Approximation and Projection (UMAP).

(Revised) *After integration, principal component analysis (PCA) was performed on the scaled*

gene expression matrix. To cluster the cells, a graph-based clustering approach built upon initial strategies developed by Macosko and colleagues was applied (Macosko et al, Cell, 2015). Briefly, a KNN graph was constructed based on 30 principal components using the FindNeighbor function. Then, cells were clustered using the FindClusters function with a resolution of 1 and visualized by Uniform Manifold Approximation and Projection (UMAP). (Line 1294-1299 of the new tracked-change MS)

Line 105 of the old MS: we deleted “widely” in MS. (Line 169 of the new tracked-change MS)

Line 110-111 of the old MS: “we observed higher proportion of high-SHM ($SHM \geq 5$) antibodies” is highlighted. We have updated this section based on our new analysis. (Line 174-178 of the new tracked-change MS)

Line 115-117 of the old MS: “These are from pooled samples- not sure if you can generalize... should qualify that it represents an average..”

Answer:

We explained the use of the pooled sample in detail in answer to reviewer 2’s 1st point, we also included the limitation of use pooled sample in the new limitation section.

Line 123: we deleted “To further investigate the antibodies induced by breakthrough infections” in MS. (Line 257 of the new tracked-change MS)

Line 133: “what about 4-4, 1-21?”

Answer:

In the original manuscript, we simply pointed out a few VH gene families with binding ratios higher than 50%, we did not mention the 4-4, 1-21 VH gene families here because the number of binding antibodies isolated for these 2 families was low. In the updated manuscript, we used the following criteria to define the VH gene families with high binding ratios: (1) VH gene families containing more than 5 mAbs; (2) proportion of binding mAbs higher than 60%. The list of VH gene families was changed from “IGHV1-46, IGHV3-21, and IGHV3-66” to “IGHV2-5 (100.00%), IGHV3-66 (66.67%), IGHV3-53 (66.67%) and IGHV3-33 (66.67%)”. (Line 269-276 of the new tracked-change MS). Recall of these mentioned VH gene families was also observed in Omicron BA.1 breakthrough infections (Kaku et al, Science Immunology, 2022).

Line133-135: “not sure what this mean”

Answer:

We apologize for the confusion, here we were trying to correlate our SHM levels of isolated antibody sequences (**Fig. 1g** of previous MS) with the “on-average” SHM level of the VDJ transcriptome data (**Fig. 1d** of previous MS). We revised the sentence as: (**Revised**) “Comparisons between the binding mAb sequences and known mAb sequences revealed the accumulation of somatic hypermutations in isolated binding mAbs (**Fig. 1g**). This is consistent to our observation that B cells from patients primarily with Delta variant breakthrough infections

show sign of increased level of somatic hypermutation by single cell V(D)J transcript data (Fig. 1d)." (Line 272-276 of the new tracked-change MS).

Line 146: We corrected the spelling of "correlating". (Line 339 of the new tracked-change MS).

Line 148: We have changed "*compromised but substantial*" to "*reduced but still substantial*" (Line 341 of the new tracked-change MS)

Line 290: "No discussion of limitation?"

We added a "Limitation of the study" section at the end of the main text in the new manuscript. (Line 654-662 of the new tracked-change MS).

Line 293: "I am not convinced that "many" binding antibodies are able to cross-react to Omicron. If I read the results correctly, the authors stated 14/110 isolated could bind to Omicron in a BINDING assay, and only 6 bound Omicron spike?"

Answer:

We changed "*many*" into "*some*". (Line 565 of the new tracked-change MS).

Line 294-295: "The paper by Cao et al compared neutralizing antibodies."

Answer:

Cao and colleagues (Cao et al, Nature, 2021) reported that over 85% of 247 neutralizing antibodies they tested are escaped by Omicron BA.1. They showed that 64 (25.91%) antibodies retain binding to Omicron BA.1 and 32 (12.96%) antibodies neutralize Omicron BA.1 with an IC50 for pseudovirus neutralization below 1 µg/mL. In our new analysis, 13/63 (22.22%) binding antibodies keep binding to Omicron BA.1 and 6 (9.52%) neutralize Omicron BA.1 with an IC50 for pseudovirus neutralization below 1 µg/mL (**Supplementary Tables 2 and 3**). Although the percentages in these two studies are close, the 247 neutralizing antibodies reported by groups worldwide may be biased towards more potent ones among others in the reported antibody isolation exercises. Therefore, the actual percentage of previously isolated antibodies which still retain binding or neutralizing to Omicron BA.1 may be lower.

We added comments: "Previously, it has been identified that only 32 of the 247 previously isolated neutralizing antibodies retained neutralization of Omicron BA.1 pseudovirus". (Line 566-568 of the new tracked-change MS).

Line 315-317: we deleted the sentence "*Likewise, repeated antigen stimulations through vaccinations and breakthrough infections likely selected affinity matured bnAbs bearing high frequencies of point mutations and indels*" in MS. (Line 599 of the new tracked-change MS).

Line 482: "Y-axes in Figure 2c YB13-292 is off by a log? Why is wild type at 1000?"

Answer:

The K_D comparisons used K_D values cited in **Extended Data Table 3**. The relative binding affinity to WT RBD by YB-258 and YB13-292 had been arbitrarily defined as 100 and 1000, this was done to facilitate easier comparison within their respective groups so that relative K_D values in each group would have integer values. To remove confusion, we have normalized the relative binding affinity to WT RBD to 1 instead in **Fig. 2c and Extended Data Fig. 10d**. We updated **Fig. 2c and Extended Data Fig. 10d**.

Line 666: "Data missing. Please provide further clarification on whether subjects received 1 or 2 doses, and at least a summary statistic of when the last dose was administered.

Unvaccinated- what kinds of symptom severity?

Also, is there data on vaccine and delta titers for the subjects?"

Answer:

We included more comprehensive patient information, including vaccine doses, symptoms, disease severity, etc. in **Extended Data Table 1**. Unfortunately, we have no data about the time of the last vaccine dose.

Line 675: "average duration of sample collection between vaccinated and non-vaccinated groups? Long hospitalization due to prolonged shedding can occur"

Answer:

We added hospitalization durations for patients in the **Extended Data Table 1**.

Line 886:"what does slash mean?? single or two doses?"

Answer:

The slash means "no information" in the old MS. We updated the information of vaccine and dose in **Extended Data Table 1**.

Reviewer #3 (Remarks to the Author):

Yu and colleagues describe in this manuscript the molecular mechanism for enhanced efficacy of affinity matured neutralizing antibodies against heterologous variant of concerns (VOC). The authors analyzed memory B cells (MBC) from Delta breakthrough infections and found that they express antibodies with higher numbers of somatic hypermutations (SHMs). They isolate several antibodies able to bind the Omicron variant BA.1 and structurally analyzed two of them. Their analysis shows how the variation introduced by the SMHs contribute to increase the antibody contact area with the epitope. The authors conclude that booster vaccines by increasing the evolution of SHMs in the pool of MBC, provide the immune system with resistant antibodies able to counteract the escape mutations present in the new VOC.

It has been reported in the literature that, with the emergence of VOC and the repeated antigen exposure of the population, the antibodies generated after vaccination or re-infection show affinity maturation and the increase in SHMs results in an increased binding to the spike protein (Muecksch F et al. Immunity 2021 - Moriyama S et al. Immunity 2021 - Sokal et al. Immunity 2021). The emergence of Omicron carrying several mutations in the RBD domain of the spike caused several breakthrough infections in the population of vaccinees. However, several authors have identified in vaccinees and infected subjects, up to 30-50% of cross-neutralizing antibodies able to bind this VOC, which are amplified by a booster vaccine eventually (Sokal et al. Immunity 2022 - Lanz et al. <https://doi.org/10.21203/rs.3.rs-1518378/v1> - Garcia-Beltran et al. Cell 2021 - Cameroni et al. Nature 2021). Some of these antibodies in complex with the spike protein have been also structurally studied by cryoelectron microscopy (Chi et al. <https://doi.org/10.1038/s41392-022-00987-z> - Sheward et al. <https://doi.org/10.1101/2022.01.03.474825>).

Structural characterization of cross-neutralizing antibodies is necessary to understand how the immune system is coping with the antigenic drift of the virus and how to design more effective vaccines. In this regard the present study is adding new knowledge to the field but it needs to be placed in the context of the published work and the obtained results need to be discussed in comparison to the mechanisms of neutralization proposed in published work.

We thank the reviewer for recognizing new information provided by this study and pointing out specific directions for manuscript improvement. We have added most of the references the reviewer had mentioned in appropriate places in the discussion section of the new manuscript.

(Line 610-618 of the new tracked-change MS).

In particular some point of the paper would need further clarification:

1) Line 96: clarify the specificity of the two numbers of B cells, are these S1+ and RBD+ respectively?

Answer:

We apologize for the confusion.

From the patient PBMC, CD19⁺ B cells were first enriched from pooled PBMCs with Miltenyi CD19 MicroBeads kit. The enriched CD19⁺ B cells were then stained with PE labelled anti-human CD27 antibody, SARS-CoV-2 biotinylated RBD protein (His Tag) conjugated with FITC-streptavidin, and biotinylated S1 protein (His Tag) conjugated with APC-streptavidin. CD19⁺CD27⁺Delta-RBD⁺Delta-S1⁺ B cells were sorted using a BD AriaFusion flow-cytometer. The purity of sorted cells was rechecked by flow-cytometry, we did not verify the antigen specificity of the sorted B cells with other methods. So, the sorted cells were denoted as CD19⁺CD27⁺Delta-RBD⁺Delta-S1⁺ specific memory B cells. We have added more details in the Methods section for clarification.

Previously described at Line 96 of the original manuscript, after removal of several patient groups, the number of single-cell transcriptome data and single-cell V(D)J data are for 3286 and 3554 B cells respectively. The reason for the B cell number difference is that for certain cell we were not able to obtain both the single-cell transcriptome data and single-cell V(D)J data after respective quality controls. We have added the following sentence in the section describing our sequencing data to further explain how the sequencing data were obtained.

“After standard quality control, from the sorted CD19⁺CD27⁺RBD⁺S1⁺ B cells we obtained single-cell transcriptome data for 3286 CD19⁺CD27⁺RBD⁺S1⁺ B cells and 3554 single-cell V(D)J data for the same CD19⁺CD27⁺RBD⁺S1⁺ B cell population.” (Line 140-161 of the new tracked-change MS).

2) Fig.1b: are these the RBD+ B cells?

Answer:

As explained above, these are CD19⁺CD27⁺Delta-RBD⁺Delta-S1⁺ B cells. We also explicitly stated which cell sequencing data were used for **Fig. 1** panels by adding sentence *“Single-cell transcriptome data (3286 cells) are used for the analyses shown in **b**, **c** and **h**. Single-cell V(D)J data (3554 cells) are used for the analyses shown in **d**, **e**, **f**, and **g**.”* (Line 834-836 of the new tracked-change MS).

3) Fig.1d: put the dashed lines in black; it is barely visible. Please add the mean value to these lines. It seems that there is no difference

between the lines in Delta breakthrough and non-vaccinated but the graph shows statistical significance.

Answer:

We thank the reviewer's suggestion. We updated **Fig. 1d** with the dashed lines changed to black and the mean values marked. After removing several pooled-samples, we found that there was no significant difference on average SHM between these two groups. However, we observed a significantly higher proportion of unmutated VH sequences for B cells from non-vaccinated group, showing elevated SHM level in the patient group primarily with Delta variant breakthrough infections. We presented the new analysis results in the updated **Fig. 1d**.

4) Fig.1i: please indicate why some of the antibodies are marked in red.

Answer:

The six antibodies were more potent cross-neutralizing antibodies and we therefore analysed them in detail and coloured them in red in the previous MS. We have added this information to the legend of **Fig. 1i**: "Six potent cross-neutralizing mAbs (IC50 < 0.05 mg/ml) were marked with *." (Line 855-856 of the new tracked-change MS).

5) Please add in the text (not only in the excel table) how many of the 110 selected antibodies are S1- and RBD-specific and how many are only S1-specific.

Answer:

We tested all the 117 antibodies by RBD binding and 53 are only RBD-specific, we selected some of the tighter RBD binder and further tested their S1 binding, we found 34 of the 53 selected antibodies are S1- and RBD-bispecific. Because we did not test all antibodies for S1 binding, so we do not want state their S1 specificities.

6) Correct the title of Table S1; it says Table S2.

Answer:

We revised the information and updated **Supplementary Tables 1-3**.

7) Line 145: Figure 2b is mentioned before figure 2a, please invert the numbering.

Answer:

We inverted the sequence of **Fig. 2a and b**.

8) Figure 2, BLI experiment: what is S-R? Please explain. Add "wildtype" to the first row of BLI data.

Answer:

S-R is a construct where we mutated the multibasic S1/S2 cleavage site PRRAR to a single R. Detailed structure characterization was performed for this construct and compared to the 2P construct in Xiong X. et. al. NSMB, 2020 (DOI: 10.1038/s41594-020-0478-5), and the data

showed that S-R spike has more closed spikes. Probably due to this property S-R spike somewhat reduces binding of various type of antibodies comparing to 2P spike (He P. et. al. Nature Microbiology, 2022, DOI:10.1038/s41564-022-01235-4). Following the reviewer's suggestion, we have changed the "S-R" designation as "S-R/Wildtype". For clarity, we also added more information in the "Protein expression and purification" section of Methods. (Line 1386-1399 of the new tracked-change MS).

9) Figure 2, neutralization graphs: please indicate that it is pseudovirus.

Answer:

We added "pseudovirus neutralization" in the **Fig. 2a** panel title.

10) Line 146: discrepancy between binding and neutralization has been reported in other studies too.

Answer:

We thank the reviewer for this suggestion. We added relevant references to the manuscript and revised the sentence as: "*It has been reported in other studies that binding activities are not always correlating with antibody virus neutralizing activities (Yuan M, et al. Science. 2020, Wec AZ, et al. Science 2020).*" (Line 343-344 of the new tracked-change MS).

11) Line 152: use "good" instead of "very good" especially in comparison to the binding of YB9-120 and YB13-208.

Answer:

We have changed "very good" to "good". (Line 346 of the new tracked-change MS).

12) Line 156: There is a difference in the resistance to the RBD mutations between the two antibodies with YB9-258 showing resistance to all the mutations tested (Fig.2c and ED Fig. 2a and b). This difference is not highlighted in the text.

Answer:

We thank the reviewer for this comment. We highlight the difference by adding the sentence "*We performed further BLI assays with RBDs of variants or bearing single substitutions to test their effect on YB9-258 and YB13-292 binding. We found that YB9-258 maintains binding to all the tested variant RBDs and RBDs with single mutations (Fig. 2c and Extended Data Fig. 2a). While YB13-292 is resistant to most single RBD mutations that were previously known to be highly detrimental to binding of many known RBD antibodies (Harvey, W. T. et al. Nat Rev Microbiol 2021) (Fig. 2c and Extended Data Fig. 2b). RBD substitutions in Kappa (L452R, E484Q) and Lambda (L452Q, F490S) variants abolish YB13-292 binding.*" (Line 350-356 of the new tracked-change MS).

13) Line 168: additional work can be cited here (Robbiani et al. Nature

2020 - Barnes et al. nature 2020).

Answer:

We have included the suggested references. (Line 389 of the new tracked-change MS).

14) Lines 173-182: please cite work reporting similar results listed in this paragraph such as: Hong et al. Nature 604, 2022 - Guo et al. Cell Reports 39, 2022 - McCallum et al. Science 375, 2022

Answer:

We have included the suggested references. (Line 403 of the new tracked-change MS).

15) Line 187: please explain the rationale for using the complex with two antibodies and cite previous work which used the same strategy (i.e., McCallum et al. Science 2022)

Answer:

The rationale was complex we were originally intended to obtain complex with higher molecular weight near the epitope to allow EM software to do focus refinement to improve resolution. However, we found simultaneous incubation with antibodies YB9-258 and R1-32 disrupted the spike and the S1 fragment in complex with 2 Fab was suitable to facilitate high resolution structure determination near the YB9-258 binding interface. We added "*In order to obtain more suitable samples for high resolution cryo-EM structure determination, after extensive efforts, following a similar strategy as previously described (McCallum et al. Science 2022),*" (Line 413-415 of the new tracked-change MS).

16) The binding affinity of YB9-258 and YB13-292 for the RBD of different VOC is different yet both antibodies neutralize Omicron BA.1. In light of the structural data the authors could discuss the molecular basis of this difference. Only the possible mechanism of neutralization is mentioned but not the possible interference with the binding to the different RBDs.

Answer:

We thank the reviewer for this suggestion. We added the following comments in sections describing structures of YB9-258 and YB13-292 epitopes respectively:

Line 429-435 of the new tracked-change MS: "In this epitope, residue 417 is centrally located while residues 484, 452 and 490 are outside of the epitope. K417N has been usually found to completely abolish binding of "class 1" VH3-53 antibodies (Yuan, M., et. al., Science 2021, Wang, R., et al., Immunity 2021). Although K417N only mildly affects YB9-258 binding but its effect is the strongest among common RBD single substitutions we tested (Fig. 2c, left panel). Consistently, YB9-258 showed reduced binding to RBDs of Beta and Omicron variants both containing the K417N substitution, while RBDs with E484K/Q, L452R and F490S substitutions had little effect on YB9-258 binding."

Line 512-516 of the new tracked-change MS: "Consistent with structural data, substitutions at 452, 484 and 490, all within the epitope, affect YB13-292 binding (Figs. 2c and 4b).

Substitutions at 417 and 478, outside of the epitope, have very little effect (Figs. 2c and 4b). However, different from 47D1, for which E484K and F490S were able to completely abolish binding, YB13-292 is able to maintain reasonable binding to RBDs with E484K and F490S substitutions (Extended Data Fig. 2b, c)."

17) Lines 293-295: Is it possible that the high number of binding antibodies isolated by this study is the result of a bias introduced by the selection criteria adopted to identify the 270 antibodies studied? The other studies were looking at an unbiased antibody repertoire.

Answer:

In fact, we found (with the latest analysis) "13/117 (~11%) Delta binding mAbs can bind to Omicron BA.1 RBD", this percentage may be quite high. Therefore, our selection criteria we adapted from Cao's method (Cao, Y., et al., Cell, 2020), may be indeed biased towards identifying cross-reactive antibodies.

18) Line 321: please cite Manner et al. 2022 here and not only in Material and Methods

Answer:

We have added this reference as ref.46. (Line 608 of the new tracked-change MS).

19) Line 553: The yellow masks mentioned in the figure legend are not shown in the ED fig.3

Answer:

We apologize for the unclarity, the masks were represented in the **Extended Data Fig. 3** but were not very obvious. We have increased the contrast of the yellow color for the masks and we high-lighted the intermediates structures with yellow boxes on which yellow masks were applied for local density extraction.

OTHER CHANGES

We updated “Omicron” in the original manuscript to “Omicron BA.1” throughout the manuscript.

For antibodies, YB9-120, YB12-197, YB13-208, YB9-258, YB13-281, and YB13-292, we investigated their binding to Omicron BA.2, BA.3 and BA.4 spikes and RBDs and these data are included in **Fig. 2b** and **Extended Data Fig. 11**.

We have deposited the single cell sequencing data (including those for the groups mentioned in the original manuscript, with the accession number PRJCA012020) in the National Genomics Data Center (<https://bigd.big.ac.cn/>), it will be validated by the curator and it should be available upon publication, the preview link is included (<https://ngdc.cncb.ac.cn/gsa-human/s/ASvvWkms>). Cryo-EM density maps and structural models have been deposited in the EMDB and PDB, the accession numbers are summarized in the structural data statistics table and they are ready to be released upon request. We included these info in the Data Availability Section. (Line 1534-1551 of the new tracked-change MS).

Line 98-99 of the new tracked-change MS: We changed the sentence there to “*the molecular mechanisms underlying such superior immune responses remained incompletely characterized.*”

Line 133-137 of the new tracked-change MS: We revised the sentence there as: “*4 pooled samples of peripheral blood mononuclear cells (PBMCs) were derived from blood samples of 15 COVID-19 convalescent patients primarily with breakthrough infections (13 have been confirmed vaccinated) of SARS-CoV-2 Delta variants (patient details are shown in **Extended Data Table 1**). Their plasmas show binding to various SARS-CoV-2 RBDs by ELISA (**Supplementary Fig. 1 and Supplementary Table 1**).*”

Line 259-262 of the new tracked-change MS: We added more information about the mAbs: “*A total of 117 candidate mAbs (with an average VH gene SHM rate of 11.88% on nucleotide level) were selected, expressed and purified. Among them, 63 (with an average VH gene SHM rate of 9.63% on nucleotide level) could bind to RBD or S1 of Delta variant*”.

Line 262-266 of the new tracked-change MS : We changed the sentence there to “*To determine whether the 63 binding mAbs developed cross-variant activity, they were further tested binding to RBDs of SARS-CoV-2 wildtype, variants of concern (VOCs), variants of interest (VOIs), and SARS-CoV-1. 22 of the 63 mAbs showed cross-binding to at least 5 of the VOCs and VOIs (**Supplementary Table 3**).*”

Line 425 of the new tracked-change MS: We changed “S1” to “*the disintegrated S1*”.

Line 642-652 of the new tracked-change MS: We have reworked the last paragraph of the Discussion section.

Line 842 of the new tracked-change MS: We moved citation for CoV-AbDab (Raybould et al, Bioinformatics, 2021) (Known mAb) to **Fig. 1** legend.

Line 1221-1234 of the new tracked-change MS: We updated the information of patients enrolled in this study (after group removal) in the Methods section.

Line 1374-1384 of the new tracked-change MS: We added the details of authentic SARS-CoV-2 neutralization assay to the Methods section.

REVIEWER COMMENTS

Reviewer #1 (Remarks to the Author):

The authors have adequately addressed my concerns

Reviewer #2 (Remarks to the Author):

The authors have done a commendable job in addressing my concerns. I believe the revised manuscript is much improved.

Reviewer #3 (Remarks to the Author):

This reviewer wishes to acknowledge the work of the authors in revising the manuscript accordingly to the reviewers' suggestions.

I have only three comments on this revised manuscript.

1) While the description of the constructs used in the study is much clearer now, it seems that only Omicron BA.1 carries the 6P mutations. It has been reported that stabilizing mutations may interfere with the overall structure of the spike protein (its open/close status or the up/down number of RBDs), so this difference in the constructs needs to be underscored when presenting comparison with other variants (for example in fig.3).

2) The rationale for using the addition of R1-32 antibody to the complex spike-YB9-258 antibody is clarified in the revised version. However, the fact that R1-32 antibody disrupt the structure of the spike is only hinted in the sentence "We obtained a 3.3 Å resolution structure of the disintegrated S1 bound to YB9-258 and R1-32". I think that this effect needs to be underscored and the interactions described for the shed S1 need to be related to S1 in the context of the spike.

3) Point 17 – lines 293-295. The authors acknowledge that their "selection criteria may be biased towards identifying cross-reactive antibodies". I think this point should be mentioned or at least listed in the limitations of the study.

Remaining reviewer comments:

Reviewer #1:

The authors have adequately addressed my concerns

Response:

We thank the reviewer's constructive comments which helped us to improve the manuscript.

Reviewer #2:

The authors have done a commendable job in addressing my concerns. I believe the revised manuscript is much improved.

Response:

We thank the reviewer's constructive comments which helped us to improve the manuscript.

Reviewer #3:

This reviewer wishes to acknowledge the work of the authors in revising the manuscript accordingly to the reviewers' suggestions.

We thank the reviewer's constructive comments which helped us to improve the manuscript.

I have only three comments on this revised manuscript.

1) While the description of the constructs used in the study is much clearer now, it seems that only Omicron BA.1 carries the 6P mutations. It has been reported that stabilizing mutations may interfere with the overall structure of the spike protein (its open/close status or the up/down number of RBDs), so this difference in the constructs needs to be underscored when presenting comparison with other variants (for example in fig.3).

We thank the reviewer's suggestion and we apologize for the confusion. We have updated our method section "protein expression and purification" (line 716-735 of the track-changed MS). Briefly, all the spikes used for structural studies are 6P stabilized (structures shown in **Figs. 3 and 4** are 6P stabilized spikes). While all spikes used for BLI are based on the S-R/Wildtype construct without any proline stabilization (**Fig. 2b**). We agree with the reviewer that proline stabilization can affect spike conformation, therefore, we used spike constructs of the same kind for comparable experiments. We hope the above info is now clarified in the revised method section "protein expression and purification". We also updated throughout the text to indicate that Omicron BA.1 S-6P and Wildtype S-6P spikes are shown for the determined structures, these changes include: several references to spike structures in the main text, labels in **Figs. 3 and 4**, cryo-EM data processing procedures (**Supplementary Figs. 4 and 5**) and cryo-EM statistics table (**Supplementary Table 6**) to clearly identify that they are 6P spikes (Omicron BA.1 S-6P and Wildtype S-6P). The purified spikes used in BLI (**Fig. 2b**) are based on S-R/Wildtype spike without proline stabilization, and the labels in **Fig. 2b** indicate that mutants are based on the S-R/Wildtype spike construct. We now explicitly state the above info in the "protein expression and purification" section. We hope this should further clarify what kind of

spike proteins used in this study.

2) The rationale for using the addition of R1-32 antibody to the complex spike-YB9-258 antibody is clarified in the revised version. However, the fact that R1-32 antibody disrupt the structure of the spike is only hinted in the sentence “We obtained a 3.3 Å resolution structure of the disintegrated S1 bound to YB9-258 and R1-32”. I think that this effect needs to be underscored and the interactions described for the shed S1 need to be related to S1 in the context of the spike.

We thank the reviewer for this suggestion, we briefly described the interactions between YB9-258, R1-32 and disintegrated S1-RBD: “In this complex, both YB9-258 and R1-32 bind to RBD surfaces that would be partially obstructed if the RBD is “down” within a spike trimer. We have previously found that antibody binding to such epitopes can promote spike opening resulting in premature triggering or disintegration of spike trimer^{10,27}” (line 282-285 of the track-changed MS). We refer to two of our previous works (Walls, A. et. al. Cell, 2019; He P. et. al. Nature Microbiology, 2022), in which we have studied antibody binding to partially obstructed spike epitopes and consequences in details. In the He P. et. al. paper, interactions between VH3-53 antibody (of which, YB9-258 is a member) and R1-32 antibody to SARS-CoV-2 RBD in the context of a whole spike were also discussed.

3) Point 17 – lines 293-295. The authors acknowledge that their “selection criteria may be biased towards identifying cross-reactive antibodies”. I think this point should be mentioned or at least listed in the limitations of the study.

We agree with the reviewer’s point and we have added “The antibody selection criteria used may be biased towards identifying cross-reactive antibodies.” in the Limitation of the study section. (Line 557-558).

Other changes

We use the present tense to describe new findings in this MS.

We updated the acknowledgement.